# PartSDF: Part-Based Implicit Neural Representation for Composite 3D Shape Parametrization and Optimization

**Nicolas Talabot**                                                    *nicolas.talabot@epfl.ch*
*EPFL*

**Olivier Clerc**
*EPFL*

**Arda Cinar Demirtas**
*Bilkent University*

**Alexis Goujon**
*Neural Concept*

**Hieu Le**                                                              *hle40@charlotte.edu*
*EPFL,*
*University of North Carolina at Charlotte*

**Doruk Oner**                                                        *doruk.oner@bilkent.edu.tr*
*Bilkent University*

**Pascal Fua**                                                            *pascal.fua@epfl.ch*
*EPFL*

**Reviewed on OpenReview:** *https://openreview.net/forum?id=zl43C1yBKv*

## Abstract

Accurate 3D shape representation is essential in engineering applications such as design, optimization, and simulation. In practice, engineering workflows require structured, part-based representations, as objects are inherently designed as assemblies of distinct components. However, most existing methods either model shapes holistically or decompose them without predefined part structures, limiting their applicability in real-world design tasks. We propose PartSDF, a supervised implicit representation framework that explicitly models composite shapes with independent, controllable parts while maintaining shape consistency. Thanks to its simple but innovative architecture, PartSDF outperforms both supervised and unsupervised baselines in reconstruction and generation tasks. We further demonstrate its effectiveness as a structured shape prior for engineering applications, enabling precise control over individual components while preserving overall coherence. Code available at https://github.com/cvlab-epfl/PartSDF.

## 1 Introduction

Engineering design is fundamentally part-driven. Whether it's a car with distinct wheels, a chair with adaptable legs, or an industrial mixer with a removable helix, objects are designed and reasoned about in terms of their components. These parts are not just geometric regions—they carry semantics and functional roles and must satisfy a number of constraints. In real workflows, engineers rarely optimize or modify a shape as a whole. Instead, they manipulate individual parts, often under precise constraints, while expecting the overall assembly to remain coherent and valid.

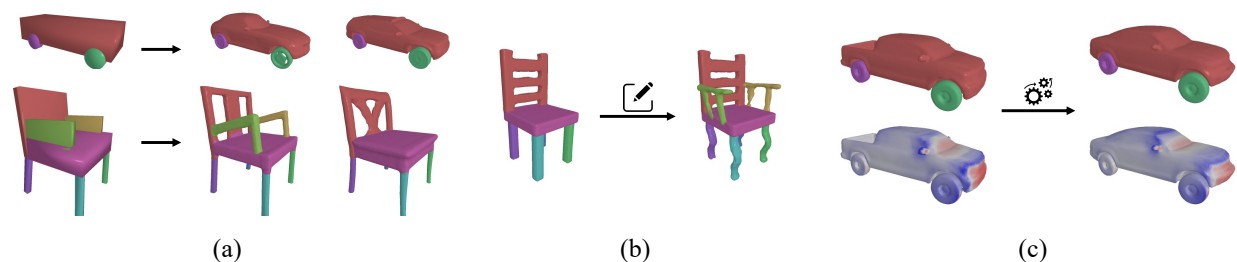

(a)             (b)             (c)

Figure 1: **Part-based implicit representation.** PartSDF is a simple and modular approach to representing composite 3D shapes for many different purposes: (a) Shape generation, possibly conditioned on a part layout. (b) Part-based manipulation. (c) Part-aware optimization. In this example, the car body, shown in red, is optimized to reduce aerodynamic drag while maintaining the shape and position of the wheels fixed.

Despite this, most 3D shape modeling methods relying on machine learning treat objects as indivisible units—holistic fields, point clouds, or meshes—with no modular structure. In particular, Implicit Neural Representations (INRs), first introduced in (Park et al., 2019; Mescheder et al., 2019; Chen & Zhang, 2019) and then refined in works such as Zhang et al. (2022; 2023), yield impressive fidelity and flexibility but often encode shapes as global functions with no built-in notion of parts. This makes tasks like part-specific manipulation, targeted optimization, or structured shape generation surprisingly difficult, often requiring cumbersome post-processing or bespoke architectures.

Although introducing parts into INRs might seem straightforward, it is, in fact, far from trivial. Modeling parts in isolation can easily break the continuity and alignment of the overall shape (Wu et al., 2020; Deng et al., 2022). Training such models often assumes clean, watertight geometries for each part—an assumption rarely met in real-world data. And even if each part is modeled successfully, how can we ensure that they remain responsive to one another—so that modifying one component, like enlarging the wheel of a car, naturally causes other parts, such as the car body, to adapt in response?

In this work, we introduce PartSDF, a simple yet effective framework for learning part-aware shape representations using signed distance functions. Each part is represented by a latent vector and a pose—capturing its geometry and spatial configuration—and decoded by a novel shared implicit model. The full shape emerges by taking the minimum across part SDFs, producing a seamless composite. This formulation is modular, supports precise part control, and naturally composes into globally coherent shapes.

To make part-based modeling practical and robust, we develop a decoder architecture that enables dynamic interaction between parts during inference. This decoder involves multiple lightweight convolutions that operate within each part and across parts respectively, enabling part-wise information exchange into a shared intermediate representation and allowing each part to adapt itself based on the geometry of others. This yields consistent deformations, without heavy computations and without resorting to complex hierarchical assemblies or attention-based systems (Deng et al., 2022; Hertz et al., 2022; Li et al., 2024). Complementing this, we propose a novel supervision scheme that bypasses the need for clean, watertight part meshes. Instead of requiring explicit SDF supervision for each part, we derive supervisory signals from the global shape's SDF, applied only within regions of space closest to each part. This strategy aligns with the natural spatial semantics of parts and allows effective training even when part segmentations are not watertight, broadening the method's applicability without compromising geometric fidelity.

This results in a versatile and fully differentiable multi-part representation that is both compact and expressive. It can be applied across tasks—including shape reconstruction, part-aware generation, and constrained optimization—without modifying the core auto-decoder architecture, as illustrated in Figure 1. Even though its architecture is simpler than that of other state-of-the-art methods, PartSDF not only improves performance on traditional benchmarks but also opens up practical capabilities that are hard to achieve with existing methods, such as optimizing a car body for drag while keeping its wheels fixed or generating diverse furniture layouts with interchangeable components.

This combination of simplicity and effectiveness is made possible by the following innovations.

- We introduce a supervised, part-aware implicit representation that models each component independently, ensuring both expressiveness and part consistency for composite objects, a key component of which is an innovative decoder that enforces consistency without undue complexity.

- We demonstrate that our approach serves as a versatile basis for various tasks such as shape reconstruction, manipulation, and optimization, all with the same core decoder network.

- We propose a novel part supervision technique relying on the full shape's SDF and inter-part losses, applicable to both watertight and non-watertight segmentations.

Through comprehensive evaluations, we show that PartSDF effectively captures the structure of composite shapes, making it well-suited for engineering applications requiring part-aware representation and control.

## 2 Related Work

Over the last several decades, the evolution of traditional CAD systems has reflected a fundamental tension between simplicity and expressiveness. Early approaches relied on basic geometric primitives such as spheres, cylinders, and NURBS surfaces (Piegl, 1991), offering mathematical precision but limited representational power. Recent years have witnessed a shift toward more sophisticated primitives, from simple cuboids (Tulsiani et al., 2017; Niu et al., 2018; Sun et al., 2019; Kluger et al., 2021) to learned deep representation, which we briefly review here. We start with generic 3D shape representations learned from data and continue with more flexible part-based approaches.

### 2.1 Learned 3D Shape Representation

Learning-based methods for 3D shape representation have evolved significantly, beginning with explicit ones such as voxel grids, point clouds, and meshes. Voxel methods (Wu et al., 2015; 2016; Choy et al., 2016b; Dai et al., 2017) partition 3D space into grids but are memory-intensive at finer resolutions, which can be mitigated using octrees (Riegler et al., 2017; Tatarchenko et al., 2017), but only up to a point. Point clouds reduce memory costs by representing shapes as a set of points but ignore connectivity, which can compromise topological consistency (Fan et al., 2017; Yang et al., 2018; Achlioptas et al., 2018; Peng et al., 2021; Zeng et al., 2022). Meshes, though well-suited for detailed surface representation, impose a rigid topology that is difficult to modify (Groueix et al., 2018; Kanazawa et al., 2018; Wang et al., 2018; Pan & Jia, 2019).

Implicit Neural Representations (INRs) offer a flexible alternative, defining shapes as continuous functions of the 3D space that encode the surface implicitly (Park et al., 2019; Mescheder et al., 2019; Chen & Zhang, 2019; Xu et al., 2019). It can then be recovered explicitly using meshing algorithms (Lorensen & Cline, 1987; Lewiner et al., 2003; Ju et al., 2002). Even though these meshing algorithms may not be themselves differentiable, differentiability can be preserved by relying on the implicit function theorem (Guillard et al., 2024), enabling back-propagation from the explicit surface, for example, when optimizing a shape to maximize its aerodynamic performance (Baqué et al., 2018). Extensions enable shape manipulation (Hao et al., 2020), point-cloud reconstruction (Peng et al., 2020), and training directly from point data (Atzmon & Lipman, 2020; Gropp et al., 2020). Newer works (Sitzmann et al., 2020; Takikawa et al., 2021) improve the accuracy further with new latent structures, such as grids (Yan et al., 2022; Mittal et al., 2022), irregular grids (Zhang et al., 2022) or unordered sets (Zhang et al., 2023), and using generative models within these latent spaces.

### 2.2 Part Based Models

As effective as they are, the INRs described above model 3D shapes as single entities, limiting their capacity to represent structured, part-based composite objects. This requires decomposing the shapes into their component parts, which can be done in a supervised or unsupervised manner.

**Unsupervised Part Decomposition.** Early unsupervised approaches learn shape abstractions using local primitives such as cuboids (Tulsiani et al., 2017; Zou et al., 2017; Sun et al., 2019; Smirnov et al., 2020; Yang & Chen, 2021), superquadrics (Paschalidou et al., 2019; 2020), anisotropic Gaussians (Genova et al., 2019), or convexes (Deng et al., 2020), approximating complex shapes as combinations of simpler parts.

More recently, complex objects are better represented by predicting and deforming primitives (Paschalidou et al., 2021; Shuai et al., 2023), creating deformable part templates (Hui et al., 2022), or performing part-based co-segmentation, which consistently divides shapes into parts across a dataset without relying on labeled boundaries (Chen et al., 2023) or does so for only a subset of the data (Chen et al., 2019). Instead, RIM-Net (Niu et al., 2022) learns a hierarchical structure of INRs, while PartNeRF (Tertikas et al., 2023) proposes a rendering-based approach for generating part-aware editable 3D shapes. SPAGHETTI (Hertz et al., 2022) predicts Gaussian parts from a global latent vector and reconstructs them into a single cohesive object, supporting interactive editing of user-defined parts. It can be combined with SALAD (Koo et al., 2023), which employs a cascaded diffusion model for part generation.

However, the parts learned by these methods are typically arbitrary and lack semantic meaning, making them unsuitable for tasks requiring specific, predefined part structures.

**Supervised Part Representation.** Supervised methods leverage known part decompositions to create explicit, structured representations of composite shapes. SDM-Net (Gao et al., 2019) proposes VAEs at the part and shape levels to learn deformable meshes. HybridSDF (Vasu et al., 2022) mixes INRs with geometric primitives to represent and manipulate shapes, while ANISE (Petrov et al., 2023) learns to assemble implicit parts for reconstruction from images and point clouds. Recently, DiffFacto (Nakayama et al., 2023) proposes cross-diffusion to generate and control part-based point clouds, which are, however, not suited to engineering needs such as simulations. Instead, PQ-NET (Wu et al., 2020), ProGRIP (Deng et al., 2022), and PASTA (Li et al., 2024) focus on implicit composite shape generation. PQ-NET uses a recurrent neural network (Cho et al., 2014) as an auto-encoder for sequences of parts, using a latentGAN (Achlioptas et al., 2018) for generation, while ProGRIP relies on shape programs to produce composite INR shapes. PASTA employs an autoregressive transformer to predict part bounding boxes that are decoded into a single global shape.

While effective for shape generation, these methods typically forgo part consistency across shapes, thus a continuous parametrization, and the representation of individual parts. Thus, there is a need for an approach that leverages part supervision to produce modular, flexible representations that can output composite shapes by parts and act as a prior in backward tasks such as optimization. PartSDF aims to fulfill that need.

**Part-Aware Generation from Images.** In addition to methods trained with 3D part supervision, several recent works explore part- or object-aware 3D generation, typically from images. Part123 (Liu et al., 2024) and PartGen (Chen et al., 2024) both begin by generating multi-view renderings with associated 2D part segmentations. Part123 then employs NeuS (Wang et al., 2021) to recover part geometries, while PartGen predicts completed part images to fill occlusions before reconstructing their 3D geometry; it can also operate from text prompts in addition to images. Concurrently, MIDI (Huang et al., 2025), addresses single-view scene reconstruction by segmenting objects and conditioning a 3D diffusion model on each object instance, with cross-object attention, while CAST (Yao et al., 2025) performs open-vocabulary scene reconstruction from an RGB image using large-scale pre-trained foundation models, diffusion, and physics-aware post-processing.

Unlike PartSDF, these approaches primarily focus on image-to-3D pipelines and emphasize image reconstruction, as opposed to geometric accuracy or ease of manipulation. In contrast, PartSDF learns supervised, part-aware implicit representations with explicit latent parametrization, enabling coherent part manipulation and constrained optimization, which are crucial in engineering and design scenarios.

## 3 Method

We introduce PartSDF, a modular and structured composite shape representation built on signed distance fields (SDFs). As shown in Figure 2(a), each object is represented by a set of parts, each one being parameterized by a latent code capturing its geometry, along with pose parameters defining its spatial placement (Section 3.1). These per-part representations are decoded into SDFs using a *cross-part* decoder (Section 3.2), which supports both independent part modeling and inter-part coordination. It is trained in an auto-decoding fashion (Park et al., 2019) in which the latent vectors and the decoder's weights are learned simultaneously, with supervision applied at both the global and part levels (Section 3.3).

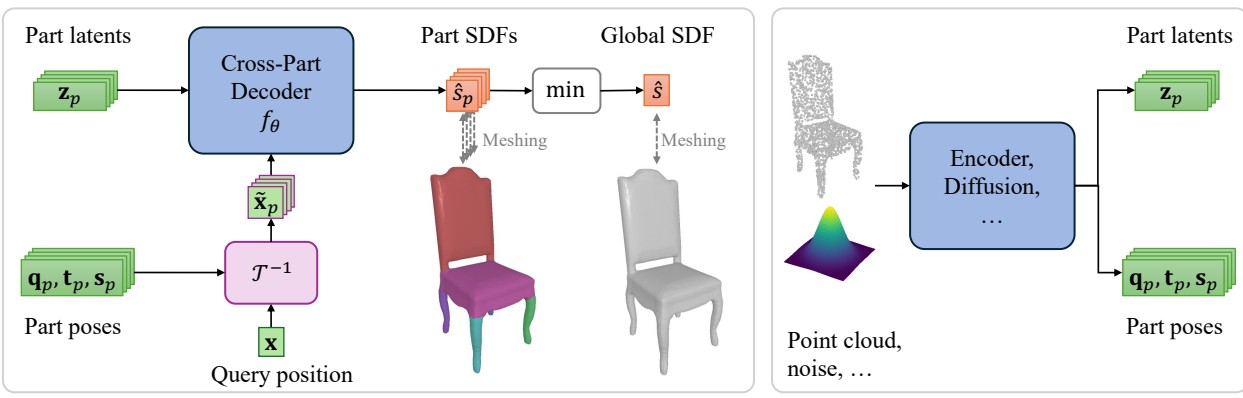

(a) Composite shape auto-decoding          (b) Part encoding or generation

Figure 2: **PartSDF pipeline.** (a) Our model's core is a part auto-decoder $f_\theta$ that takes as input part latents $\mathbf{z}_p$ and poses expressed in terms of a quaternion $\mathbf{q}_p$, translation $\mathbf{t}_p$, and scale $\mathbf{s}_p$, along with the query position $\mathbf{x} \in \mathbb{R}^3$. It outputs signed distances $\hat{s}_p$ for all parts at the queried position, which may be combined into the global signed distance. (b) A secondary model may be used based on the task at hand, such as encoders to map a given modality, *e.g.*, point clouds, to part latents and poses, or a diffusion model to generate them from noise.

As shown in Figure 2(b), our architecture supports secondary models for *part encoding or generation*, allowing further adaptation of part latents and poses for tasks such as shape reconstruction or generation, while maintaining the same core part decoder for efficient inference and manipulation (Section 3.4).

## 3.1 SDF Computation

In our composite shape representation, each part is described by a latent vector $\mathbf{z}_p \in \mathbb{R}^Z$ and a pose $\mathbf{p}_p \in \mathbb{R}^{10}$, consisting of a rotation quaternion $\mathbf{q}_p \in \mathbb{R}^4$, translation $\mathbf{t}_p \in \mathbb{R}^3$, and scale $\mathbf{s}_p \in \mathbb{R}^3$. Given a query point $\mathbf{x} \in \mathbb{R}^3$, it is mapped into each part's canonical space via inverse transformation $\mathcal{T}^{-1}$, resulting in the transformed query points

$$\hat{\mathbf{x}}_p = \mathcal{T}^{-1}(\mathbf{x}, \mathbf{p}_p) = \mathbf{R}_p(\mathbf{x} - \mathbf{t}_p)/\mathbf{s}_p, \ \forall p \ , \tag{1}$$

where $\mathbf{R}_p$ is the rotation matrix obtained from $\mathbf{q}_p$. Our cross-part decoder $f_\theta$, with $\theta$ containing all trainable parameters, operates on all parts' latents $\mathbf{Z}$ and transformed query points $\hat{\mathbf{X}}$ to output the part SDFs as

$$\hat{\mathbf{s}} = f_\theta\left(\mathbf{Z}, \hat{\mathbf{X}}\right). \tag{2}$$

The SDFs for all parts can then be combined to recover the full shape representation as $\hat{s} = \min_p \hat{s}_p$.

## 3.2 Cross-Part Auto-Decoder

To represent complex shapes in a modular yet coherent manner, we introduce an innovative decoder that explicitly handles multiple interacting parts. At each layer, the decoder maintains a multi-part feature matrix $\mathbf{X}^l \in \mathbb{R}^{P \times D_l}$, where each row corresponds to a part and each column to a shared feature dimension. This structure is maintained throughout the network, preserving an explicit notion of parts across all layers. As a result, the decoder's output naturally yields one part per row—enabling direct, part-specific SDF predictions.

To achieve this, we design the network to alternate between two types of layers: single-part layers that update parts independently and cross-part layers that enable controlled feature sharing across parts.

*Single-part layer.* The first type of layer processes each part separately, allowing the network to learn the shape distribution of individual parts, denoted as $h_{\text{sp}}$:

$$\mathbf{x}_p^{l+1} = h_{\text{sp}}^l(\mathbf{z}_p, \mathbf{x}_p^l) = \sigma\left(\mathbf{W}^l \mathbf{x}_p^l + \mathbf{b}^l + \mathbf{W}_z^l \mathbf{z}_p + \mathbf{b}_p^l\right) \ , \tag{3}$$

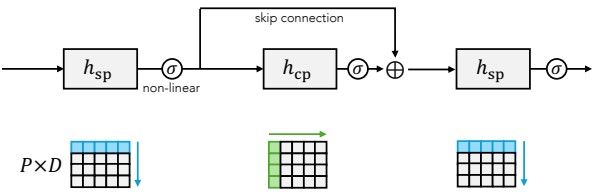

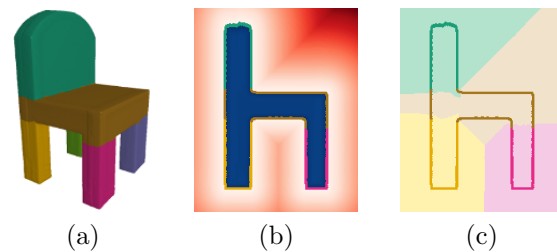

(a)  (b)  (c)

Figure 3: **Cross-part adaptation in PartSDF.** Our decoder alternates between updating each part independently and sharing information across parts, allowing them to adapt to one another while preserving modularity. This is implemented through a sequence of lightweight convolutions applied along rows and columns of the part feature matrix.

Figure 4: **Supervision for non-watertight parts.** (a) Semantic parts of a chair, (b) a 2D slice of its signed distances (red/blue) with parts highlighted in color, and (c) the specific regions of space where each part is supervised. This enables training without requiring parts to be watertight.

where $\sigma$ is an activation function, $\mathbf{x}_p^l$ denotes the $p$-th row of $\mathbf{X}^l$, *i.e.*, the features of part $p$ at layer $l$, $(\mathbf{W}^l, \mathbf{b}^l)$ are the layer's parameters, $\mathbf{W}_z^l$ is the parameters of a *latent modulation* (Dupont et al., 2022), and $\mathbf{b}_p^l$ the learnable bias of part $p$ at layer $l$.

*Cross-part layer.* The second type of layer enables feature aggregation across parts, denoted as $h_{\mathrm{cp}}$. For each feature dimension $d$, we write

$$\tilde{\mathbf{x}}_d^{l+1} = h_{\mathrm{cp}}^l(\tilde{\mathbf{x}}_d^l) = \sigma\left(\tilde{\mathbf{W}}^l \tilde{\mathbf{x}}_d^l + \tilde{\mathbf{b}}^l\right) \ , \tag{4}$$

where $\tilde{\mathbf{x}}^l$ denotes the $d$-th column of $\mathbf{X}^l$, *i.e.*, the $d$-th feature dimension across all parts, and $(\tilde{\mathbf{W}}^l, \tilde{\mathbf{b}}^l)$ are the layer's parameters. This layer enables the decoder to capture inter-part dependencies and improve adaptability during part manipulation with as little as $P^2 + P$ added trainable parameters per such layer, with $P$ the number of parts, as shown in our ablation study of Section 4.5. We also add a skip connection to ensure these layers don't mix part identities.

By stacking alternating $h_{\mathrm{sp}}$ and $h_{\mathrm{cp}}$ layers (see Figure 3), which can be efficiently implemented using convolutions, the decoder enables each part to specialize while remaining aware of its context—producing modular, adaptive, and coherent shape representations. We provide network details in Appendix B.1.

### 3.3 Learning from Global Shape Supervision

Let us consider a dataset $\mathcal{D}$ where each shape $\mathcal{S}$ is decomposed into up to $P$ parts, given as the segmentation of its surface, as commonly found in online databases. To establish initial poses for each part, we fit simple primitives—cuboids or cylinders—to the segmented parts and use these primitives' poses as the part poses. Therefore, each element of the dataset becomes a tuple $\mathcal{D} = \left\{(\mathcal{S}, \{\mathcal{P}_p\}, \{\mathbf{p}_p\})_i\right\}$ of the shape, its parts, and their poses $\mathbf{p}_p = (\mathbf{q}_p, \mathbf{t}_p, \mathbf{s}_p)$.

**Training the Decoder.** For watertight shapes $\mathcal{S}$, PartSDF's decoder is then trained in an auto-decoding fashion (Park et al., 2019), where model parameters $\theta$ and part latent vectors $\mathbf{z}_p$ are optimized jointly. Note that we can directly use the parts poses from $\mathcal{D}$ during training. When a part is missing from a specific shape, we assign it the average pose of that part over the dataset, allowing the model to naturally learn to output SDF $> 0$ for nonexistent parts, without having to predict *existence scores*, as in Petrov et al. (2023).

Some parts $\mathcal{P}_p$, however, may *not* be watertight as they correspond to open surface fragments rather than closed volumes—a common characteristic of online part-based CAD models. This makes it difficult to compute reliable per-part signed distance fields, especially near part boundaries or in regions where parts overlap. While one option is to repair or approximate each part's volume (Wu et al., 2020; Petrov et al., 2023), this often introduces artifacts and scaling issues. Instead, we take advantage of a key observation: when reconstructing the global shape as the minimum across all part SDFs, only the part whose surface is closest to a given point affects the output. We therefore supervise each part using the global shape's SDF, but only in regions of space where that part is closest to the surface.

Formally, a part $\mathcal{P}_p$ is supervised only at points

$$\mathbf{x} \in \left\{ \mathbf{x} \in \mathbb{R}^3 \mid p = \arg\min_i d_i(\mathbf{x}) \right\}, \tag{5}$$

where $d_i(\mathbf{x})$ is the projection distance of $\mathbf{x}$ to part $i$, as depicted by Figure 4. This region-based supervision ensures that each part learns only the portion of space it is responsible for and avoids penalizing it in regions where it is occluded or irrelevant. Crucially, it allows us to train directly from surface-based part annotations, without requiring watertight meshes or explicitly constructing per-part SDFs.

To further encourage modularity, we introduce a non-intersection loss that penalizes overlapping negative SDF predictions across parts. This promotes clean spatial separation, ensuring that each part occupies a distinct region of space—crucial for enabling localized editing, part replacement, and consistent composition under the minimum-SDF fusion.

**Loss Function.** For any shape from $\mathcal{D}$, we minimize

$$\mathcal{L} = \mathcal{L}_{\text{sdf}} + \mathcal{L}_{\text{part}} + \mathcal{L}_{\text{inter}} + \lambda \sum_p \|\mathbf{z}_p\|^2, \tag{6}$$

where $\lambda$ is a hyperparameter controlling the latent $L2$-regularization and the loss terms are defined below.

For the whole shape, we minimize

$$\mathcal{L}_{\text{sdf}} = \frac{1}{|\mathcal{X}|} \sum_{(\mathbf{x}_i, s_i) \in \mathcal{X}} |\hat{s} - s_i|, \tag{7}$$

where $\mathcal{X}$ is a set of sampled points in 3D and around $\mathcal{S}$, with their ground truth SDF $s_i$ from the full shape $\mathcal{S}$. $\hat{s}$ is the predicted SDF at those points. For notational simplicity, we omit the dependency on $\theta$ and the $\mathbf{z}_p$ along with the clamping of SDF values (Park et al., 2019).

For individual parts, we minimize

$$\mathcal{L}_{\text{part}} = \frac{1}{|\mathcal{X}|} \sum_{(\mathbf{x}_i, s_i, p_i) \in \mathcal{X}} |\hat{s}_{p_i} - s_i|, \tag{8}$$

with an additional index $p_i$ of the part closest to $\mathbf{x}_i$, re-using $\mathcal{X}$ as a slight abuse of notation to explicitly state that we compute these losses on the same 3D samples. $\hat{s}_{p_i}$ is the predicted part SDF at those points.

Finally, we introduce a non-intersection loss that pushes the SDF of parts to be positive at each position $\mathbf{x}_i$ where at least two parts have SDF$< 0$. Using $\hat{\mathcal{X}}$ as the subset of such 3D points, the loss is written

$$\mathcal{L}_{\text{inter}} = \frac{1}{|\hat{\mathcal{X}}|} \sum_{\mathbf{x}_i \in \hat{\mathcal{X}}} |\mathbf{w}_i \cdot \hat{\mathbf{s}}_i|, \tag{9}$$

where $\hat{\mathbf{s}}_i$ is the vector of predicted part SDF at point $\mathbf{x}_i$ and $\mathbf{w}_i = \text{softmax}(\tilde{\mathbf{s}}_i)$, defining $\tilde{\mathbf{s}}_i$ as $\hat{\mathbf{s}}_i$ with all positive values replaced by $-\infty$. This pushes towards $> 0$ more strongly the SDFs that are closer to it.

### 3.4 Part Encoding and Generation

Once PartSDF's decoder is trained, it is frozen to be used for downstream tasks, optionally in conjunction with separate specialized networks. While this requires training two networks independently, it makes the training of each one easier. Furthermore, the decoder need only be trained once and used again and again for the different tasks.

**Encoding** To reconstruct a shape from a given modality, *e.g.*, point clouds or images, encoders can be trained to directly predict part latents and poses, to then be decoded with PartSDF: With input data $\mathcal{I}$ corresponding to the shape of our training data $\mathcal{D}$, encoders are trained to map this new modality to the part latents and poses of our pre-trained decoder. As an example, we show how to reconstruct unseen shapes for which part decomposition is unknown in Section 4.1. We do this by training a point cloud encoder to predict part latents and poses, which are then refined in an auto-decoding strategy and part-agnostic manner.

**Generation**  For shape generation with a coherent part set, generative models can be trained to map random noise to the parts parametrization. Generative Adversarial Networks (GAN) (Goodfellow et al., 2014) have been used to generate shape latents, often dubbed *latentGANs* in related work (Achlioptas et al., 2018; Chen & Zhang, 2019) such as PQ-NET (Wu et al., 2020). Recently, diffusion models (Sohl-Dickstein et al., 2015; Ho et al., 2020) have gained a lot of traction for their generative performances, even in shape (Zhang et al., 2023) or part-based generation (Koo et al., 2023). Generating new shapes with PartSDF can be achieved by training such generative models on the part poses and latents of our pre-trained decoder and, even if not shown here, could be conditioned on images or texts (Zhang et al., 2023; Koo et al., 2023).

## 4 Experiments

We demonstrate our method on several tasks and compare it to multiple part-based baselines. Additional implementation details and results are given in Appendices B and G.

**Datasets**  Public shape datasets like ShapeNet (Chang et al., 2015) and PartNet (Mo et al., 2019) often suffer from over-segmentation or inconsistent part definitions. To address this, we use three curated datasets with clean, consistent part decompositions: (1) *Car*, a hand-processed ShapeNet subset with separated wheels; (2) *Mixer*, liquid mixers with a helix, tube, and two attach points (Vasu et al., 2022); and (3) *Chair*, a cleaned PartNet-based set with individually segmented legs and arms. These datasets contain 1046, 1949, and 1332 shapes with 5, 4, and 8 parts, respectively. We use 80% for training and 20% for testing.

**Baselines.**  We compare against part-based methods across different supervision levels to evaluate their ability to represent composite shapes: DAE-NET (Chen et al., 2023) (unsupervised deformable parts), BAE-NET (Chen et al., 2019) (weakly supervised with 8 labeled shapes), PQ-NET (Wu et al., 2020) and PASTA (Li et al., 2024) (fully supervised, though PASTA does not output parts). We also evaluate 3DShape2VecSet (Zhang et al., 2023), a state-of-the-art, non-part-based method that, while unsuitable for composite shapes, provides an upper bound on achievable accuracy without enforcing part structure.

**Metrics.**  Reconstruction accuracy is assessed using three different metrics: Chamfer-Distance (CD) for surface accuracy, Intersection over Union (IoU) for volume, and Image Consistency (IC) (Guillard et al., 2022) for shape appearance and normals. Part reconstruction is evaluated by averaging the per-part IoU and as we do not make the assumption that parts must be watertight, we compute part occupancies using the same strategy as for our training losses in Section 3.3. For generation, we report Minimum Matching Distance (MMD) and Coverage Score (COV) (Achlioptas et al., 2018), using CD as the distance metric.

### 4.1 Shape and Part Reconstruction

We evaluate the accuracy of shape and part reconstruction across all datasets and methods. At inference, our model uses auto-decoding: the decoder remains frozen while latent vectors are optimized to minimize reconstruction loss. We also report results for PartSDF using a single part and refer to it as Ours-1P. Meshes are reconstructed using Marching Cubes (Lewiner et al., 2003) at a resolution of 256. We report quantitative results in Table 1, with qualitative ones shown in Figure 5. Across all metrics, our method achieves the best results on cars and chairs and equivalent results on mixers to Ours-1P, which does not reconstruct individual parts. Notably, 3DShape2VecSet struggles with the mixer's helical structures, likely due to its positional encoding's difficulty with periodic geometry. Other part-based methods perform significantly worse on both surface and volume metrics.

One thing that handicaps DAE-NET, BAE-NET, and PQ-NET is their reliance on voxelized data despite being INR-based: For speed and memory, the data is binarized and voxelized at $64^3$, which delivers efficient and fast models but greatly limits accuracy. This is particularly visible in the case of the thin helix of the mixers or the equally thin chair parts: they tend to either disappear or be inflated. Instead, PartSDF is trained with samples directly obtained from the original shape $\mathcal{S}$ and yields superior accuracy, capturing detailed part-specific geometry and structure. While the more recent PASTA is also trained with these samples and improves on the other works, it still performs less well than our approach. Because it only uses the part bounding boxes as input, it may fail to capture the correct local details and, thus, fail to properly

Table 1: **Shape reconstruction.** We compute the average Chamfer-Distance CD ($\times 10^4$), Intersection over Union IoU (%), and Image Consistency IC between reconstructions and corresponding test shapes. We also report average per-part Intersection over Union pIoU (%) for part-based methods. Ours-1P uses a single part to reconstruct the full shape and Ours-PC uses a point cloud encoder to get initial part latents and poses that are refined in a part-agnostic manner.

| | CD ($\downarrow$) | | | IoU (%, $\uparrow$) | | | IC ($\uparrow$) | | | pIoU (%, $\uparrow$) | | |
| --- | --- | --- | --- | --- | --- | --- | --- | --- | --- | --- | --- | --- |
| | Car | Mixer | Chair | Car | Mixer | Chair | Car | Mixer | Chair | Car | Mixer | Chair |
| 3DShape2VecSet | 2.73 | 5.19 | 3.35 | 92.03 | 68.82 | 92.13 | 0.887 | 0.851 | 0.909 | - | - | - |
| Ours-1P | 1.37 | **1.44** | 3.74 | 97.41 | **95.16** | 93.78 | 0.924 | **0.978** | 0.913 | - | - | - |
| DAE-NET | 28.38 | 15.93 | 100.49 | 81.91 | 70.14 | 56.37 | 0.797 | 0.918 | 0.662 | 34.71 | 31.83 | 52.80 |
| BAE-NET | 33.43 | 13.68 | 106.60 | 75.17 | 33.38 | 40.79 | 0.771 | 0.818 | 0.594 | 21.83 | 31.14 | 38.72 |
| PQ-NET | 30.20 | 17.75 | 38.90 | 72.00 | 26.71 | 49.09 | 0.754 | 0.739 | 0.690 | 36.27 | 25.67 | 51.48 |
| PASTA | 12.15 | 6.27 | 40.89 | 83.88 | 50.83 | 60.36 | 0.855 | 0.942 | 0.750 | - | - | - |
| Ours | **1.27** | 1.60 | **1.30** | **98.02** | 94.43 | **97.17** | **0.931** | **0.978** | **0.942** | **94.89** | **90.42** | **93.67** |
| Ours-PC | 1.28 | 1.73 | 1.57 | 97.95 | 83.38 | 97.09 | 0.930 | 0.966 | **0.942** | 94.25 | 78.61 | 88.13 |

Figure 5: **Shape reconstruction.** Reconstruction of test shapes with all models. For part-based methods, we color each part with a different color and translate the helix outside of the mixers for visualization.

reconstruct some shapes, as in the chairs of Figure 5. It is also unable to represent individual parts, despite being part-aware.

Table 2: **Shape generation.** We compute the Minimum Matching Distance MMD ($\times 10^4$) and Coverage Score (COV) (%), using the Chamfer Distance (CD), between generated and test shapes.

| | MMD-CD ($\downarrow$) | | | COV-CD (%, $\uparrow$) | | |
| | Car | Mixer | Chair | Car | Mixer | Chair |
|---|---|---|---|---|---|---|
| 3DShape2VecSet | **20.57** | 13.59 | **54.96** | 83.81 | 39.23 | **88.39** |
| PQ-Net | 30.61 | 25.25 | 70.68 | 43.81 | 50.26 | 74.91 |
| PASTA | 27.72 | 16.31 | 118.35 | 40.95 | 51.03 | 23.97 |
| Ours | 20.67 | **11.97** | 55.46 | **85.71** | **86.67** | 83.90 |

Figure 6: **Shape generation.** We generate shapes on all datasets, and we also provide examples of pose-conditioned generation (Ours†) where part latents are generated based on the poses' coarse description of the shape (left image for each pair). When possible, we translate the helix outside of the mixer for visualization.

Additionally, to illustrate that part encoder models can be used in conjunction with PartSDF, we also reconstruct test shapes using part-agnostic point clouds: We use a point cloud encoder based on a simplified 3DShape2VecSet without final cross-attention. It predicts initial part latents and poses, refined via auto-decoding without $\mathcal{L}_{\text{part}}$. As shown in the last row of Table 1, this approach successfully recovers both global shape and parts, though very thin features remain more challenging.

## 4.2 Shape Generation

We compare PartSDF's shape generation abilities against those of 3DShape2VecSet, which relies on a diffusion model to generate its set of latent vectors, PQ-Net, which trains a latentGAN (Achlioptas et al., 2018) to yield a global latent vector, and PASTA, which uses an autoregressive transformer to generate part bounding boxes. For PartSDF, we leverage SALAD (Koo et al., 2023), a cascaded diffusion model that first generates part poses and then part latents conditioned on these poses. This also enables us to generate shapes fitting specified pose decompositions, something that PQ-Net cannot do with its single latent space, and to generate various geometries for a given pose decomposition, which PASTA cannot do as it only relies on part poses for reconstruction. We report MMD and COV metrics in Table 2 and show generated examples

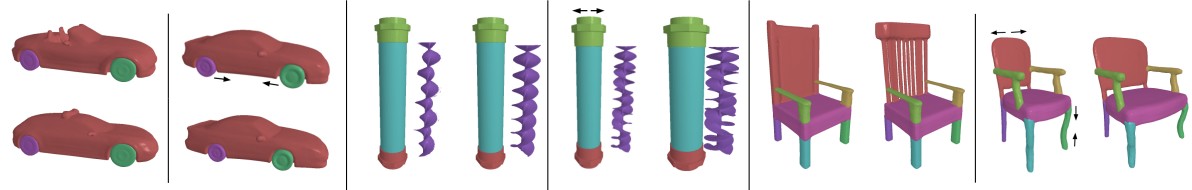

Figure 7: **Part manipulation.** We manipulate two shapes per dataset, first by changing the latent of specific parts (car body, mixer helix, and chair backrest) and second by editing part poses (car wheels, mixer width, chair width and height). In all cases, the parts adapt to the modifications and to each other, maintaining a coherent whole with the new parts.

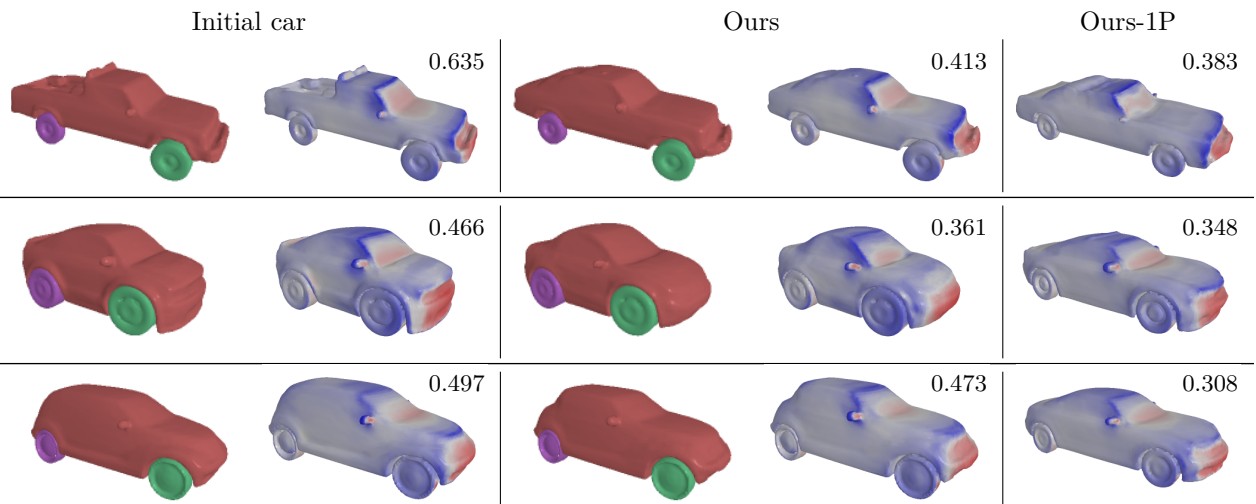

Figure 8: **Part-based optimization.** We refine the car bodies to reduce aerodynamic drag *while preserving the size and position of the wheels.* (Ours) Our part-based representation enables this. (Ours-1P) In contrast when using a single-part, everything changes, including the wheels. We highlight individual parts on the left, the surface pressure on the right (from blue to red), and give the drag coefficient $C_d$ as an inset.

in Figure 6. PartSDF achieves consistently better results than the part-aware baselines, with more detailed composite shapes. We note that despite significant integration effort on our side, the PASTA-generated poses exhibit low diversity and occasional pose errors. Our results are also better than, or on par with, those of 3DShape2VecSet, in addition that our models are part-aware, whereas those of 3DShape2VecSet are not, and therefore not usable for engineering design. Furthermore, it requires training different auto-encoder models between shape reconstruction and generation, while PartSDF uses the same decoder for both.

## 4.3 Part Manipulation

We evaluate our model's ability to perform part-specific shape manipulation by editing part latent vectors and poses in Figure 7. Both latents and poses can be edited conjointly in PartSDF; we do so separately here for visualization purposes. When modifying part latents, the appearance of the corresponding parts is changed, but the shape's structure and parts layout remain fixed. On the other hand, when changing poses, the part layout adapts, and their general appearance is unchanged. In all cases, the resulting composite shape preserves its overall consistency while fitting the editions. These qualitative results emphasize the flexibility of our part-based representation, showing that our model can adapt individual parts independently while preserving overall shape integrity.

## 4.4 Part Optimization

To demonstrate the effectiveness of PartSDF as a part-aware shape prior for downstream tasks, we address a key engineering problem: Refining the shape of a car to reduce the drag induced by air flowing over its

Table 3: **Ablation on PartSDF components.** We ablate the use of part poses, latent modulation (latMod), and cross-part layers ($h_{\mathrm{cp}}$) on the Car dataset. Best performance is achieved with poses and latent modulation, though we note that $h_{\mathrm{cp}}$ layers decrease slightly the part accuracy of our model. However, this is counterbalanced by enabling inter-part adaptability; see Figure 9.

| | CD ($\downarrow$) | IoU ($\%$, $\uparrow$) | IC ($\uparrow$) | pIoU ($\%$, $\uparrow$) |
|---|---|---|---|---|
| w/o poses | 1.43 | 97.81 | 0.928 | 91.71 |
| w/o latMod | 1.39 | 97.37 | 0.923 | 92.33 |
| w/o $h_{\mathrm{cp}}$ | **1.27** | **98.02** | **0.932** | **96.37** |
| PartSDF | **1.27** | **98.02** | 0.931 | 94.89 |

surface $\mathcal{S}$, *without* editing the wheels. Drag can be computed as the surface integral of the air pressure:

$$\mathrm{drag}_p(\mathcal{S}) = \oiint_{\mathcal{S}} -n_x(\mathbf{x}) \cdot p(\mathbf{x}) \, d\mathcal{S}(\mathbf{x}) \,, \tag{10}$$

where $p(\mathbf{x})$ is the pressure and $\mathbf{n}(\mathbf{x})$ the surface normal at point $\mathbf{x}$, with $n_x$ its component along the $x$ axis, which is directed along the car from front to back. This value is then normalized to get the drag coefficient $C_d$ (Munson et al., 2013). For simplicity, we ignore the *friction* drag that tends to be negligible for cars. To compute a *differentiable* estimate of the surface pressure $\hat{p}(\mathbf{x})$ and resulting drag, we use a GCNN surrogate model (Baqué et al., 2018). More specifically, we use a GraphSage Convolution GNN (Hamilton et al., 2017; Bonnet et al., 2022) to predict $\hat{p}(\mathbf{x})$ on the mesh's surface, using simulation data obtained with OpenFOAM (OpenCFD).

Shape optimization can then be achieved by minimizing

$$\mathcal{L}_{\mathrm{drag}} = C_d \left( \mathrm{MC}(f_\theta, \mathbf{Z}, \mathbf{P}) \right) + \mathcal{L}_{\mathrm{reg}} \left( \mathbf{Z}, \mathbf{P} \right) \,, \tag{11}$$

with respect to $(\mathbf{Z}, \mathbf{P})$, where $\mathcal{L}_{\mathrm{reg}}$ are optional regularization terms and MC is the Marching Cubes algorithm (Lewiner et al., 2003). Importantly, gradients can be computed through the meshing step (Remelli et al., 2020). In practice, we minimize $\mathcal{L}_{\mathrm{drag}}$ over the PartSDF latents $\mathbf{Z}$ and pose parameters $\mathbf{P}$, starting from their values for an existing car. In the example of Fig. 8, we optimize the shapes of several cars while keeping their wheels unchanged, which is something that our part-based approach allows and can not be done with a global representation such as the one of Ours-1P or 3DShape2VecSet (Zhang et al., 2023). While they may yield a lower final drag, the shapes do not respect design constraints, which is undesirable. For instance, in the second and last row of Figure 8, results with Ours-1P tend to converge to an average car, ignoring the specificity of the initial cars and their wheels. We provide more details and results in Appendix D.

### 4.5 Ablation Study

We conduct an ablation study to evaluate the impact of cross-part layers $h_{\mathrm{cp}}$, the use of pose parameters, and latent modulation in PartSDF, and report metrics on the Car dataset in Table 3. Pose parameters allow independent control of part transformations, which improves the shape reconstruction performance but also allows more precise part manipulation. Latent modulation, *i.e.*, injecting the part latent vectors $\mathbf{z}_p$ at every $h_{\mathrm{sp}}$ layer, is a simple modification that improves surface and part reconstruction, as measured by Chamfer-Distance and part IoU. On the other hand, cross-part layers $h_{\mathrm{cp}}$ cause a slight decrease in part accuracy, largely counterbalanced by the inter-part adaptability it enables. In Figure 9, we show that adding these layers to PartSDF maintains the consistency between parts when doing part manipulation. We also show the necessity of the non-intersection loss $\mathcal{L}_{\mathrm{inter}}$: without it, the losses do not prevent the parts from bleeding into one another, causing the model to learn an incorrect part decomposition with overlapping parts.

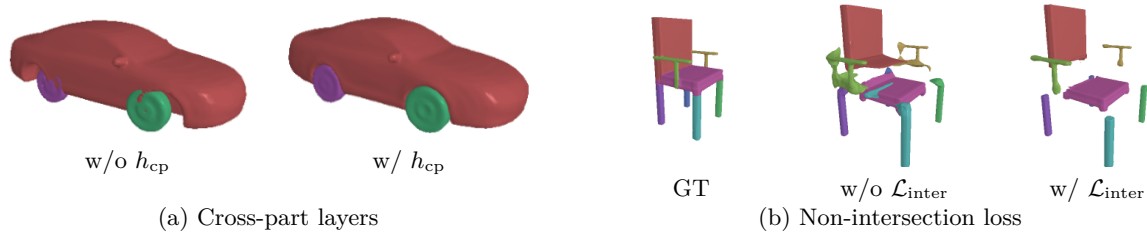

(a) Cross-part layers        (b) Non-intersection loss

Figure 9: **Ablation study of PartSDF.** (a) Without the cross-part layers $h_{cp}$ (Section 3.2), manipulating a part would not affect the others: The translated and scaled wheels incorrectly collide with the car. With these layers, manipulating some parts correctly affects the others, and the car adapts to the new wheels without intersection. (b) Without the non-intersection loss $\mathcal{L}_{inter}$, the parts incorrectly overlap. With it, the correct decomposition is learned. We visualize an exploded view of the predicted parts to see the overlaps.

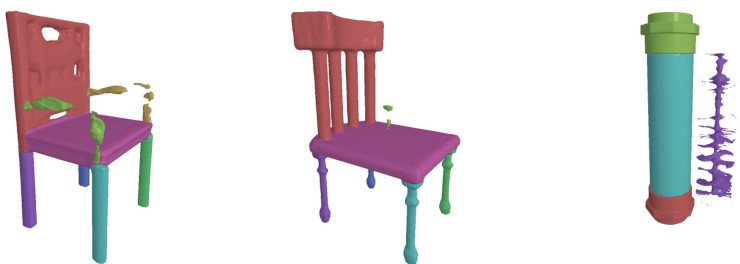

Figure 10: **Failures cases of PartSDF.** (Left) Interpolation between shapes where a part exists in one but not the other may produce floating artifacts for intermediate shapes (e.g., chair armrests). (Middle) In rare test cases, parts that should be absent can persist as small surface fragments, such as the green armrest patches, which affects Chamfer-Distance. (Right) Very thin structures, such as the helix of some mixers, remain challenging to represent and mesh accurately using an SDF-based approach.

## 5 Limitations

The main limitation of PartSDF is its reliance on part labels during training. For our intended application, computer-aided design, this is not a major issue because shapes are typically constructed from individual parts; hence, the decomposition of shapes into parts is known *a priori*. Thus, PartSDF is particularly relevant in scenarios where objects are naturally created with part-aware structures. While this limits PartSDF's applicability to datasets without predefined decompositions—such as most of those available online—it better matches with practical engineering requirements. Nonetheless, if generalization to unlabeled data is desired, PartSDF could be combined with co-segmentation methods to generate pseudo-labels for its training. PartSDF also has a few typical failure modes that are illustrated in Fig. 10. These include occasional artifacts when interpolating between shapes with different part existences, residual small surfaces for absent parts, and difficulties representing very thin structures. Such issues occur infrequently and can often be mitigated by discarding known-absent parts or by using higher-resolution meshing for thin geometries. Additionally, because the computational and memory costs scale with the number of parts, extending PartSDF to shapes with high part counts might require strategies to improve efficiency, such as sparse cross-part interactions. Finally, while we devised a strategy to handle non-watertight parts, it still assumes that they are volumetric and can be represented by an SDF. Thus, parts that are pure surfaces, such as car hoods, would not be naturally handled without inflating them by wrapping a volume around them. An interesting direction for future work is to use *unsigned* distance functions to solve this.

## 6 Conclusion

In this work, we introduced PartSDF, a modular and supervised approach specifically designed for composite shape representation. PartSDF enables flexible, part-based shape modeling, supporting independent part

manipulation and optimization while maintaining overall shape coherence. Our method leverages a simple architecture, achieving strong performance across tasks such as shape reconstruction, manipulation, and generation. Experimental results demonstrate that PartSDF consistently outperforms baseline methods in part-level accuracy and can be adapted to a wide range of tasks, highlighting its effectiveness as a robust shape prior for composite structures. This flexibility makes PartSDF particularly suited for applications in fields like engineering design, where precise control over individual components is essential for customization and optimization. While PartSDF achieves promising results, future work will explore enhancing part interactions to further support applications involving highly dynamic shapes or complex inter-part dependencies. Furthermore, developing more advanced topological supervision for each part should also be investigated to help prevent topologically inconsistent predictions, further improving its robustness and applicability.

### Acknowledgements

This work was funded in part by the Swiss National Science Foundation.

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

# A    Table of Contents

# B    Implementation Details

In this section, we describe the architecture of PartSDF (Appendix B.1), its training procedure (Appendix B.2), and the point cloud encoder and diffusion models we used (Appendix B.3). Then, we detail our datasets and their processing (Appendix B.4), the baselines against which we compare and any hyperparameter choices (Appendix B.5), and finally the metrics used in this work and how they were computed (Appendix B.6).

## B.1    Network Architecture

The final architecture of PartSDF, as presented in this work, is as follow:

The dimensionality of the part latent space is $Z = 256$, such that $\mathbf{z}_p \in \mathbb{R}^{256} \ \forall p$, and the hidden size of the single-part layers $h_{\mathrm{sp}}$ is 512, with the exception of the model trained on the *Chair* dataset where it is 256. This is because there are more parts, thus we expect each part to be simpler and it helps to alleviate the memory increase. We use weight normalization (Salimans & Kingma, 2016) on these layers, as in (Park et al., 2019). PartSDF has 8 $h_{\mathrm{sp}}$ layers, counting the input and output ones, with a cross-part layer $h_{\mathrm{cp}}$ between each pair, as described in Section 3.2. All layers, except the output one, are followed by ReLU non-linearities. Note that each $h_{\mathrm{cp}}$ has only $P^2 + P$ trainable parameters, *e.g.*, $25 + 5 = 30$ parameters in the case of cars.

## B.2    Training Procedure

The decoder model of PartSDF is trained for 2000 epochs using a batch size of 16 and 8192 sampled points per shape. We use the Adam optimizer (Kingma & Ba, 2015) with default parameters in PyTorch (Paszke et al., 2017), setting a learning rate of $5 \times 10^{-4}$ for the model and $1 \times 10^{-3}$ for the latent vectors. The learning rates are reduced by a factor of 0.35 at 80% and 90% of the total training epochs. The latent regularization weight is set to $\lambda = 10^{-4}$, and the non-intersection loss uses a softmax temperature parameter set to 0.02.

To supervise the SDF, we follow the sampling strategy used in DeepSDF (Park et al., 2019): 95% of the sampled points are near the surface, and 5% are uniformly distributed in space. The SDF values are computed

using the IGL library (Jacobson et al., 2018), assuming watertight meshes. During training, the SDF values are clamped between $-0.1$ and $0.1$ to focus the capacity of the network around the surface.

Trainings are conducted on a single NVIDIA V100 GPU with 32GB of memory, taking 1–2 days to train depending on the dataset size.

### B.3 Encoder and Diffusion Models

**Point Cloud Encoder.** In Section 4.1, we use a point encoder to map point clouds of the full shapes to PartSDF parameter space, *i.e.* part latents and poses. For this, we leverage the model of 3DShape2VecSet (Zhang et al., 2023) using their official implementation[1]. We use a smaller version of the model with 8 self-attention layers (against 24 in the original work), and without the final cross-attention layer with query positions (see Fig. 3 in their work) as we want to output part latents and poses instead of implicit values. We use learnable queries instead of point queries (see Fig. 4(a) in (Zhang et al., 2023)) to encode the point cloud, which creates an order on the latent set, allowing us to recover the parameters of specific parts. Note that we concatenate latent and pose for every part, such that the encoder outputs them both in a single vector.

**Diffusion Model.** In order to generate composite shapes with PartSDF, we use a diffusion model to generate the part latents and poses (Section 4.2), more specifically SALAD (Koo et al., 2023) with their official implementation[2]. For the second diffusion on the latents, we however make use of a transformer decoder instead of encoder so that each part latent can attend to all part poses, and the part latents are normalized before training. This last point was found crucial for the diffusion as our latent vectors tend to have significantly lower norms than random vectors sampled from a multivariate Gaussian distribution. Indeed, as in (Park et al., 2019), latent vectors are initialized such that $\mathbb{E}_{\mathbf{z}_p}\|\mathbf{z}_p\| = 1$, while Gaussian noise in $d$-dimensions has $\mathbb{E}_{\mathbf{x}\sim\mathcal{N}^d}\|\mathbf{x}\| = \sqrt{d}$.

### B.4 Datasets

For all datasets, we preprocess the meshes by centering them using their bounding box and rescaling them into the cube $[-0.9, 0.9]^3$, ensuring the longest edge of the bounding box measures 1.8. This leaves a margin around the shape to prevent the surface from leaving the meshing region, defined as the $[-1, 1]^3$ cube.

To create the SDF supervision, we perform two types of sampling (Park et al., 2019). First, we sample points on the surface of the mesh and perturb them with Gaussian noise, using variances of 0.005 and 0.0005 to generate points near the surface. Second, we uniformly sample points within the $[-1, 1]^3$ cube. The SDF values for all sampled points are computed using *libigl* (Jacobson et al., 2018). The samples and their SDF are mixed in a 95/5% ratio (near-surface to uniform samples) and stored for training. During training, a subset of these precomputed points is used, randomized at each epoch.

Additionally, each part, given as a subset of the original mesh, is fitted with a cuboid or cylinder. The orientation, translation, and scale of the primitive is then used as that part's pose.

As explained in Section 4, obtaining high-quality composite shape data is challenging, with online data usually containing non-watertight meshes, or without part decomposition or an inconsistent one. Therefore, we use three curated datasets in this work.

**Cars.** In order to perform Computational Fluid Dynamics (CFD) simulations on the cars, see Appendix D below, we need them to be correctly closed and physically plausible, *i.e.* wheels and body correctly separated. We therefore hand-process a subset of ShapeNet (Chang et al., 2015)'s cars: The wheels and body are manually selected and separated, and then remeshed to become watertight (Wang et al., 2022).

**Mixers.** This dataset was proposed by (Vasu et al., 2022) and consists in liquid mixers made of a central tube, an interior helix, and two attach points. It has the advantage of having an available, and consistent,

---

[1] https://github.com/1zb/3DShape2VecSet
[2] https://github.com/KAIST-Visual-AI-Group/SALAD

part decomposition, which is suitable to learn a composite shape representation. The main challenge lies in the thinness of the tubes and helices, and the latter's complexity.

**Chairs.** We use a clean subset of ShapeNet's chairs with semantic segmentation provided by PartNet (Mo et al., 2019), with seat, backrest, armrest, and leg classes. To enable SDF computation, we first make the chairs watertight (Stutz & A.Geiger, 2018). Then, the PartNet segmentation, originally provided as a point cloud, is transferred to the mesh using a voting scheme: each face is assigned the most frequent class among its closest points. To refine this segmentation, we further divide the armrests and legs into individual parts by detecting connected components. Chairs that fail this finer segmentation are removed to ensure a consistency within our dataset. Additionally, we discard shapes at intermediate steps if they exhibit issues such as misalignment with the segmentation point cloud or a high Chamfer-Distance relative to the original shape.

Each dataset is split into train, validation and test sets (60%/20%/20%). Model hyperparameters are tuned using the train and validation sets, and final models, as reported in the main text, are trained on train+validation and evaluated on the test set. Part poses are obtained by fitting each part with a cuboid or a cylinder.

## B.5 Baselines

**DAE-Net.** DAE-Net (Chen et al., 2023) is a recent method that learns an unsupervised part decomposition to perform co-segmentation. It creates multiple part templates that are deformed to reconstruct a shape given its voxelization as input, allowing the original shape to be segmented based on the predicted decomposition. As mentioned in our main text, unsupervised methods like DAE-Net are typically designed to discover new shape decompositions, which is unsuitable for engineering design where parts have specific predefined meanings and uses. Additionally, as shown in our experiments, the reliance on voxelized inputs significantly limits its representation ability. We use the official implementation[3] and conduct a grid search to tune the sparsity loss weight $\gamma$ (see Eqs. (5) and (6) in their work). The optimal values found were $\gamma = 0.001$ for cars and mixers, and $\gamma = 0.002$ for chairs, consistent with their reported results.

**BAE-Net.** Similarly, BAE-Net (Chen et al., 2019) encodes a voxelized input to generate part implicit fields, which can be used for co-segmentation. The key difference, and the reason for comparison to our approach, is that BAE-Net also introduces a weak supervision scheme where ground truth segmentations are provided for a small subset of the training shapes (typically ranging from 1 to 8 shapes). In this setting, the model propagates the learned decomposition across the entire dataset. Similar to DAE-Net, BAE-Net suffers from the limitations of voxelized inputs, which restrict its reconstruction accuracy. However, as shown in Figure 5, the weak supervision allows it to recover the desired decomposition in most cases, although certain parts, such as car wheels, are sometimes segmented together rather than individually. We use the official implementation[4] to train the models. For the weak supervision, we experiment with 1, 2, 3, and 8 shots, finding that 8-shot supervision consistently yields the best results. Therefore, we use these models.

**PQ-Net.** As a fully supervised baseline, we compare against PQ-Net (Wu et al., 2020). PQ-Net employs a two-stage training process: first, an implicit auto-encoder is trained to reconstruct all individual parts, followed by a GRU-based RNN (Cho et al., 2014) that learns to auto-encode sequences of parts, represented by their latent vectors and bounding box parameters. For shape generation, latentGANs (Achlioptas et al., 2018) are used to sample in the RNN's latent space. Like the previous baselines, PQ-Net relies on voxelized data and often reconstructs "inflated" shapes. While this helps against thinner parts, such as mixer helices and chair legs, the overall reconstruction accuracy remains low. Additionally, backpropagating through RNNs is notoriously difficult (Pascanu et al., 2013), and the non-continuous nature of its sequence generation makes it less suitable as a shape prior for optimization tasks. We train PQ-Net using the author's official implementation[5].

---

[3]`https://github.com/czq142857/DAE-Net`
[4]`https://github.com/czq142857/BAE-NET`
[5]`https://github.com/ChrisWu1997/PQ-NET`

Table 4: **Model sizes.** Size of the different models in term of trainable parameters. *: Most parameters are in the MLPs of the Self-Attention layers between the latent vectors. †: We only consider the parameters of the *blending network* and ignore the *object generator*, which is only relevant for generation.

|  | # of parameters |
|---|---|
| 3DShape2VecSet* | 106.1M |
| Ours-1P | 2.5M |
| DAE-Net | 3.1M |
| BAE-Net | 5.3M |
| PQ-Net | 12.8M |
| PASTA† | 9.0M |
| Ours | 2.5M |

**PASTA.** Li et al. (2024) proposes another supervised method mostly aimed at generating shape in a part-aware manner. It is made of two parts. First, an autoregressive transformer (Vaswani et al., 2017) is trained to output a sequence of part, each represented through their bounding box (or equivalently, their pose). Secondly, a transformer decoder performs a cross attention between these part bounding boxes and the query points to output an occupancy value for each, using a small final MLP. PASTA's decoder only rely on the part layout, as described by their respective bounding boxes, and as such entangles the part geometry to them. It is unable to output varying geometry without affecting the part poses. Additionally, PASTA outputs a single occupancy per shape and do not produce individual part. We use the official implementation[6] to train PASTA on our data.

**3DShape2VecSet.** In order to evaluate the reconstruction and generation quality of our shapes against a more powerful baseline, we also compare to a state-of-the-art part-agnostic method, namely 3DShape2VecSet (Zhang et al., 2023). The core of the architecture is a large auto-encoder transformer (Vaswani et al., 2017) model that encodes a point cloud into an *unordered set* of latent vectors, which are then decoded at queried positions to obtain occupancy values. For generation, a KL-divergence loss is added during training to make the auto-encoder variational and help a second-stage diffusion model. This diffusion model is trained to generate *sets* of latent vectors, thus sampling 3DShape2VecSet's latent space. As noted in the work, the models are very heavy, requiring multiple GPUs for training. Using the author's official implementation[7], we train auto-encoders *without* the KL-divergence loss for reconstruction experiments and *with* the KL-divergence for the generation experiments (as reference, we use the *same* PartSDF decoder for both experiments). We use the provided hyper-parameters.

### B.5.1 Model size comparison

We compare the models' size in term of their number of trainable parameters. As can be seen in Table 4, PartSDF is the smallest model, with 3DShape2VecSet being $\sim 40\times$ larger. In practice, 3DShape2VecSet also requires multi-GPU training, as noted in its paper, while PartSDF can be trained on a single GPU.

To understand the good performance of PartSDF in spite of its smaller size, we note that 3DShape2VecSet is an *auto-encoding* method designed for point cloud reconstruction and diffusion-based generation, not for per-shape optimization. In contrast, PartSDF is an *auto-decoding* framework that is well-suited for optimization and manipulation tasks. For completeness, we also experiment with optimizing 3DShape2VecSet latents at inference time on the *Car* dataset, with two settings: "+AD", where latents are randomly initialized and optimized, as in PartSDF, and "+AE+AD", where latents are first obtained from the point-cloud encoder and then further optimized. We report the results in Table 5.

In the +AD setting, 3DShape2VecSet struggles to optimize its latent vectors and the reconstructions exhibit bumpy surfaces and floaters, hence the lower IC and poor CD. This is consistent with observations made in

---

[6]`https://github.com/Vincent-Li-9701/PASTA`
[7]`https://github.com/1zb/3DShape2VecSet`

Table 5: **Auto-decoding with 3DShape2VecSet.** Reconstruction results when using *auto-decoding* with 3DShape2VecSet with randomly initialized latents (+AD) or encoded from a point-cloud (+AE+AD).

| | CD ($\downarrow$) | IoU (%, $\uparrow$) | IC ($\uparrow$) | pIoU (%, $\uparrow$) |
|---|---|---|---|---|
| 3DS2VS | 2.73 | 92.03 | 0.887 | - |
| 3DS2VS+AD | 57.97 | 92.49 | 0.848 | - |
| 3DS2VS+AE+AD | 1.15 | 98.78 | 0.942 | - |
| Ours-1P | 1.37 | 97.41 | 0.924 | - |
| Ours | 1.28 | 97.95 | 0.930 | 94.25 |

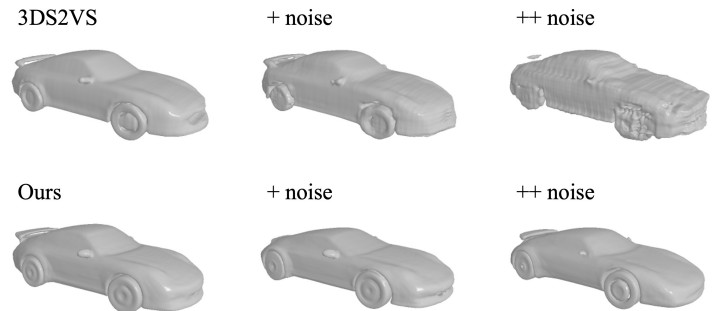

Figure 11: **Reconstruction with noisy latents.** When gradually adding gaussian noise to the latent vectors in 3DShape2VecSet and PartSDF, we observe a degradation of the shape with the former. With the latter, the output remains a slightly different, but valid, car. The noise is taken to be 5% (+noise) and 7.5% (++noise) of the latents average norm.

the DeepSDF paper's supplementary material (Sec. E, Fig. 19). This highlights that its latent space is not well suited for direct optimization, one of the main reasons we did not adopt a VecSet-style latent space for PartSDF. With +AE+AD, performance improves and slightly surpasses that of PartSDF in reconstruction accuracy.

However, 3DShape2VecSet is not a good fit for the applications we are targeting: It does not provide explicit part representations and its latent space is not designed for controllability or per-part optimization. To demonstrate this, we add small amounts of Gaussian noise to the latent vectors of 3DShape2VecSet and PartSDF before reconstruction. In Figure 11, the standard deviation of the noise is taken to be 5% and 7.5% of the latents average norm. Adding such small amounts of noise to 3DShape2VecSet's latent vectors quickly degrades the shape, while PartSDF's output remains a valid car. In other words, PartSDF's latent space is better behaved and, thus, better suited for shape optimization.

### B.6 Metrics

**Reconstruction.** As we mention in the main text, to assess reconstruction accuracy of the 3D shapes, we use three metrics: For surface accuracy, we use the Chamfer-Distance (CD), which measures the *symmetric* distance between sets of points, $\mathcal{X} = \{\mathbf{x}_i\}_{i=1}^N$ and $\mathcal{Y} = \{\mathbf{y}_i\}_{i=1}^M$, sampled from the surface of the ground truth and reconstructed surfaces respectively. Their CD is then written

$$\text{CD}(\mathcal{X}, \mathcal{Y}) = \frac{1}{N} \sum_{\mathbf{x} \in \mathcal{X}} \min_{\mathbf{y} \in \mathcal{Y}} \|\mathbf{x} - \mathbf{y}\|^2 + \frac{1}{M} \sum_{\mathbf{y} \in \mathcal{Y}} \min_{\mathbf{x} \in \mathcal{X}} \|\mathbf{y} - \mathbf{x}\|^2. \tag{12}$$

In this work, we use $N = M = 30'000$, unless specified otherwise.

To evaluate the volume accuracy, we use Intersection over Union (IoU). It is defined as the ratio of volume between the intersection of the shapes over their union. In practice, we compute it using occupancy values

on a grid: We find the smallest bounding box that contains the two shapes, then sample points on a $128^3$ grid. For each point $\mathbf{x}$, we get the binary occupancy of each shape as $o_{\mathcal{S}_1}(\mathbf{x})$ and $o_{\mathcal{S}_2}(\mathbf{x})$, where $o_{\mathcal{S}}(\mathbf{x})$ is 1 if $\mathbf{x}$ is inside the shape and 0 otherwise. The IoU is then computed as

$$\text{IoU}(\mathcal{S}_1, \mathcal{S}_2) = \frac{\sum_{\mathbf{x}} o_{\mathcal{S}_1}(\mathbf{x}) \cdot o_{\mathcal{S}_2}(\mathbf{x})}{\sum_{\mathbf{x}} \max\left(1, o_{\mathcal{S}_1}(\mathbf{x}) + o_{\mathcal{S}_2}(\mathbf{x})\right)}. \tag{13}$$

Finally, we use Image Consistency (IC) (Guillard et al., 2022) to evaluate the appearance and surface normals, using renderings of the shapes. Let $\mathcal{K}$ be a set of 8 cameras located at the vertices of a cuboid that encompasses the shape, looking at its centroid. For each camera $k \in \mathcal{K}$, we render the binary silhouettes $S_k \in \{0,1\}^{256 \times 256}$ and normal map $N_k \in \mathbb{R}^{256 \times 256 \times 3}$ of shape $\mathcal{S}_1$, and similarly $\tilde{S}_k$ and $\tilde{N}_k$ for shape $\mathcal{S}_2$. The IC between these shapes is defined as

$$\text{IC}(\mathcal{S}_1, \mathcal{S}_2) = \frac{1}{|K|} \sum_{k \in K} \text{IoU}_{\text{2D}}(S_k, \tilde{S}_k) * \text{COS}(N_k, \tilde{N}_k), \tag{14}$$

where $\text{IoU}_{\text{2D}}$ is the Intersection over Union between binary images and COS is the average cosine similarity between two normal maps. We refer interested readers to (Guillard et al., 2022) for more details.

We additionally report per-part average IoU (pIoU) as a way to measure part reconstruction accuracy, noting that CD and IC might have issues if the ground-truth parts are not watertight. In order to compute the occupancy of an open part, we rely on a strategy similar to our SDF supervision, see Figure 4. First, the occupancy of the full shape is computed on the grid. Then, for each point where the occupancy is 1, we look for the closest part to it. This part will get an occupancy of 1 at that point and all the others will get 0. Therefore, part occupancy can be seen as the intersection of the full shape's occupancy with the part's "closest region" as visualized in Figure 4(b). Note that for baselines without part correspondences, *i.e.* DAE-NET and PQ-NET, we must first match the reconstructed parts to the ground truth (GT) ones: For each shape and each reconstructed part, we match it with the GT part with which it has the highest IoU.

**Generation.** To evaluate the plausibility and diversity of the generated shapes, as compared to a held out test set, we report the Minimum Matching Distance (MMD) and Coverage Score (COV) (Achlioptas et al., 2018) respectively, using the Chamfer-Distance as the (pseudo-)distance metric.

MMD is the average distance between each shape from the test set $T$ and the generated set $G$. It is written as

$$\text{MMD}(G, T) = \frac{1}{|T|} \sum_{\mathcal{X} \in T} \min_{\mathcal{Y} \in G} \text{CD}(\mathcal{X}, \mathcal{Y}), \tag{15}$$

where we make the abuse of notation that $\text{CD}(\mathcal{X}, \mathcal{Y})$ means the Chamfer Distance between samples on the surfaces of $\mathcal{X}$ and $\mathcal{Y}$.

On the other hand, COV measures the fraction of shapes in the test set that are recovered with a greedy matching of generated shapes: each shape in G is matched with the closest shape in T. Rigorously, it is computed as

$$\text{COV}(G, T) = \frac{|\{\arg\min_{\mathcal{X} \in T} \text{CD}(\mathcal{X}, \mathcal{Y}), \forall \mathcal{Y} \in G\}|}{|T|}. \tag{16}$$

For both the baseline and our method, we generate 2000 shapes, and use 2048 surface samples to compute the CD, for efficiency concern.

## C Design Analysis

**Number of parts.** We evaluate the influence of the number of parts on the *Chair* dataset by artificially fusing or splitting parts to vary the decomposition. We report the results in Table 6 and visualize examples of parts decomposition in Figure 12. Increasing the number of parts generally improves reconstruction metrics, as simpler per-part geometries are easier to represent. However, we observe a slight Chamfer-Distance degradation due to a few failures in removing armrests during reconstruction, which suggests that

Table 6: **Varying the number of parts.** Reconstruction results on the *Chair* dataset with PartSDF when varying the number of parts in the decomposition.

| # of parts | CD (↓) | IoU (%, ↑) | IC (↑) | pIoU (%, ↑) |
|---|---|---|---|---|
| 1 | 3.74 | 93.78 | 0.913 | - |
| 2 | 1.61 | 95.91 | 0.931 | 93.08 |
| 3 | 2.00 | 96.55 | 0.936 | 93.24 |
| 4 | 1.27 | 97.13 | 0.943 | 93.69 |
| 5 | 1.51 | 97.23 | 0.943 | 93.63 |
| 7 | 1.32 | 97.21 | 0.944 | 93.12 |
| 8 | 1.30 | 97.17 | 0.942 | 93.67 |
| 9 | 1.39 | 97.37 | 0.945 | 93.22 |
| 12 | 1.76 | 97.48 | 0.945 | 93.01 |

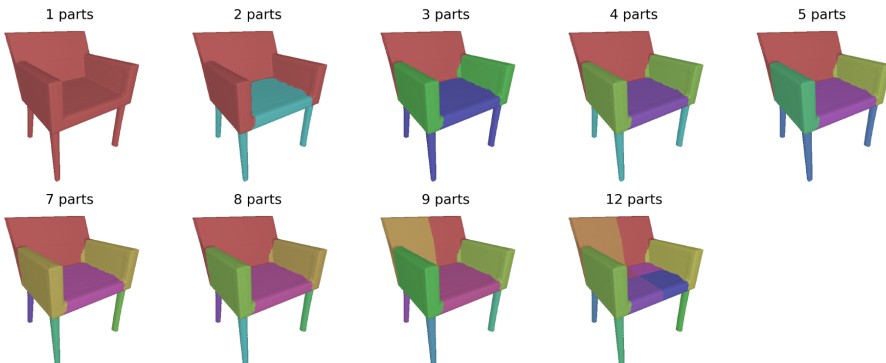

Figure 12: **Part decompositions for the *Chairs*.** We show an example of shape with its parts colored differently to visualize the part decompositions.

larger part counts may require more data and training to robustly capture cross-part interactions. Part IoU remains relatively stable, with only minor variation ($\simeq 0.6\%$). Overall, while finer decompositions can improve expressiveness, they may reduce controllability by requiring more poses to be specified and sufficient training data to model all part relationships.

**Architectural variants.** We experiment with a transformer-based architecture (Vaswani et al., 2017) to replace our cross-part decoder with self-attention, see Table 7. While feasible, this increases parameter count (from ∼2.5M to 4.8M), memory (traing samples reduced by a factor 4 to avoid out-of-memory errors) and training cost (from 2000 to 4000 epochs with lower learning-rate) for the same reconstruction accuracy. We also observed greater parts entanglement in this architecture: displacing the car wheels also affect the car length even when its scale is kept constant, see Figure 13 where the car front and back move with the wheels instead of simply adapting their wells around them. We believe this can be solved, for example by adding some shape mixing during training, but we note that our simpler cross-part decoder already enables it at a lower memory and computation cost.

**SDF aggregation strategies.** The merger of two SDFs into a single one is performed using the min operator, see, *e.g.*, `https://iquilezles.org/articles/distfunctions/`, which simply aggregates the parts. We tried replacing the min by a softmin variant and found that it performs similarly, if marginally worse, as shown in Table 7 for the *Car* dataset.

**Scalability of PartSDF.** The computational cost of PartSDF depends on both the number of parts $P$ and the feature dimension $D$. The single-part layers $h_{\text{sp}}$ scale as $O(PD^2)$, while the cross-part layers $h_{\text{cp}}$ scale as $O(P^2D)$. In our experiments, $D \in \{256, 512\}$ and $P \ll D$, so runtime and memory scale approximately linearly with the number of parts. However, for very large $P$, the combined complexity becomes $O(PD^2 + P^2D)$, similar to the cost of transformer layers (considering self-attention + MLP).

Table 7: **Comparing architectures.** Reconstruction results on the *Car* dataset using a Transformer-based decoder (Transf.) and a softmin SDF-aggregation strategy with our decoder (softmin) against PartSDF.

|         | CD ($\downarrow$) | IoU (%, $\uparrow$) | IC ($\uparrow$) | pIoU (%, $\uparrow$) |
|---------|---------|---------|---------|---------|
| Transf. | 1.25 | 98.00 | 0.931 | 93.44 |
| softmin | 1.31 | 97.90 | 0.930 | 94.78 |
| Ours    | 1.27 | 98.02 | 0.931 | 94.89 |

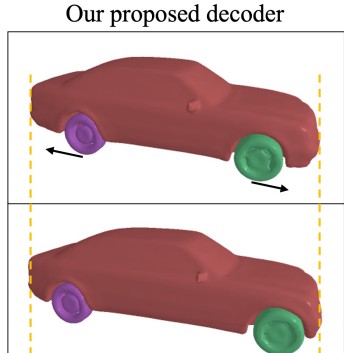 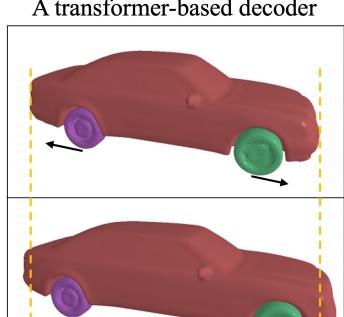

Figure 13: **Manipulation with different architectures.** We compare manipulation between a Transformer-based variant against our cross-part decoder, by translating the wheels while keeping the body's scale fixed. For the Transformer-based decoder, the length of the car is incorrectly affected.

Although our layers remain more lightweight, modeling highly complex objects with many parts would require both large $D$ and $P$ to capture fine geometric details, which may limit scalability. Extending our work to such cases is possible but will require strategies such as positional encoding, hierarchical part grouping, or sparse cross-part interactions to improve efficiency.

# D    Part Optimization

As described in Section 4.4, we optimize shapes to demonstrate how PartSDF is used as a part-aware shape prior. In this section, we provide further experimental details and results.

**Experimental Setup.** The goal of this experiment is to optimize the shape of a car to minimize its aerodynamic drag, expressed as the drag coefficient ($C_d$), by modifying its main body under the constraints that wheels cannot be modified. Direct simulation of aerodynamic performance, such as Computational Fluid Dynamics (CFD) (OpenCFD), is computationally expensive and non-differentiable without adjoint solvers (Allaire, 2015). To address this, we use a surrogate model to approximate drag-related metrics efficiently and differentiably (Baqué et al., 2018; Remelli et al., 2020).

We use a graph convolutional neural network (GCNN), specifically with GraphSage layers (Hamilton et al., 2017), to predict the surface pressure distribution of a car. The predicted surface pressure is integrated to compute the drag force (Equation (10)), which is then normalized by the mass density of the fluid, its velocity squared, and the frontal surface area of the shape to obtain $C_d$. We simulate the cars in our dataset, following the well-established setup of the DrivAer model (Wieser et al., 2014), to compute the corresponding surface pressures and $C_d$ values. The dataset thus created is used to trained the GCNN-based surrogate model.

Following this, all model parameters, those of PartSDF and the surrogate's, are frozen for the shape optimization. It is performed by adjusting the latents **Z** and/or poses **P** of the parts, starting from the initial parametrization of an existing car. In this experiment, the wheels are kept fixed and only the body is opti-

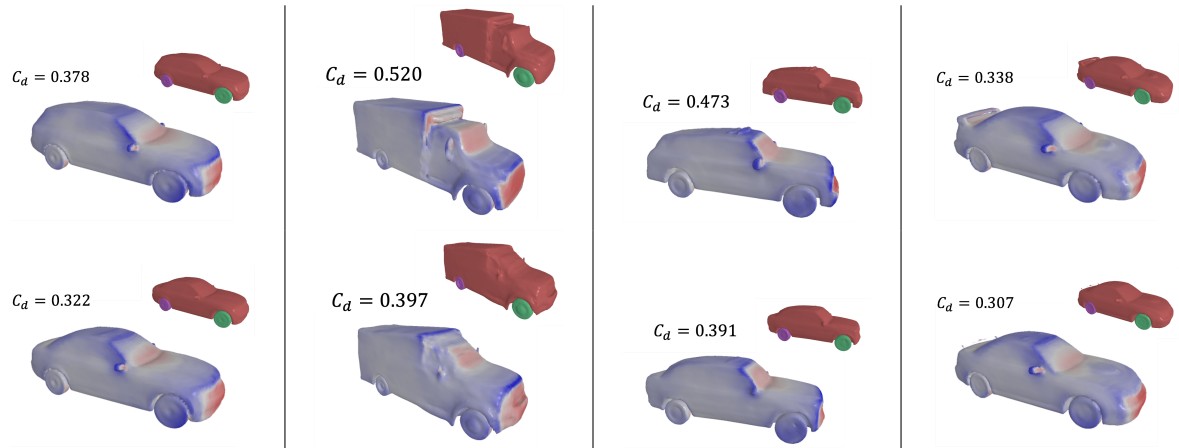

Figure 14: **Additional shape optimization.** The aerodynamic drag of the car is minimized with respect to the latent vector of the car's body, with fixed wheels. We show the initial car pre-optimization (*top*) and the resutling car optimization (*bottom*), colored by surface pressure from low to high pressure (blue to red). We also visualize the parts in the insert. We report the drag coefficient $C_d$ before and after the shape optimization, as computed by a physical simulator (OpenCFD).

mized, which is only possible because of the composite shape representation of our model that also ensures coherence between the parts.

The optimization minimizes the integrated pressure drag predicted by the surrogate model, backpropagating through the meshing to adjust the part parameters (Equation (11)). Regularization terms $\mathcal{L}_{\text{reg}}$ are applied to maintain consistency in the latent space and prevent excessive deviations:

- A $k$-nearest neighbors (kNN) loss on the latent vectors to ensure they remain close to the training distribution, as introduced in (Remelli et al., 2020), and

- an $L_2$ regularization on the deviation of latent vectors and poses from their initial values, *e.g.*, $\|\mathbf{Z} - \mathbf{Z}_{init}\|_2^2$.

To summarize, during a single iteration of this optimization process:

1. Current part latents and poses $(\mathbf{Z}, \mathbf{P})$ are used to compute the car's SDF on a grid, as required by the Marching Cubes algorithm.

2. Marching Cubes is applied on the SDF grid to obtain a mesh of the car surface $\mathcal{S} = \text{MC}(f_\theta, \mathbf{Z}, \mathbf{P})$.

3. The mesh is passed through the GCNN surrogate to obtain the surface pressure $\hat{p}(\mathcal{S})$, from which the drag coefficient $C_d$ is computed.

4. The gradient of $C_d + \mathcal{L}_{\text{reg}}$ is computed with respect to $(\mathbf{Z}, \mathbf{P})$, which are then updated through stochastic gradient descent.

This setup allows us to efficiently optimize composite shapes for aerodynamic performance while leveraging the flexibility of PartSDF's framework.

**Results.** With this experimental setup, we optimize multiple cars from our dataset and simulate the resulting shapes in OpenFOAM (OpenCFD) to obtain their final drag coefficient $C_d$. Because simulations are expensive ($> 10$h depending on the meshing resolution of the 3D space), we must limit ourselves to a subset of 35 cars. The average drag coefficients before and after optimizations are

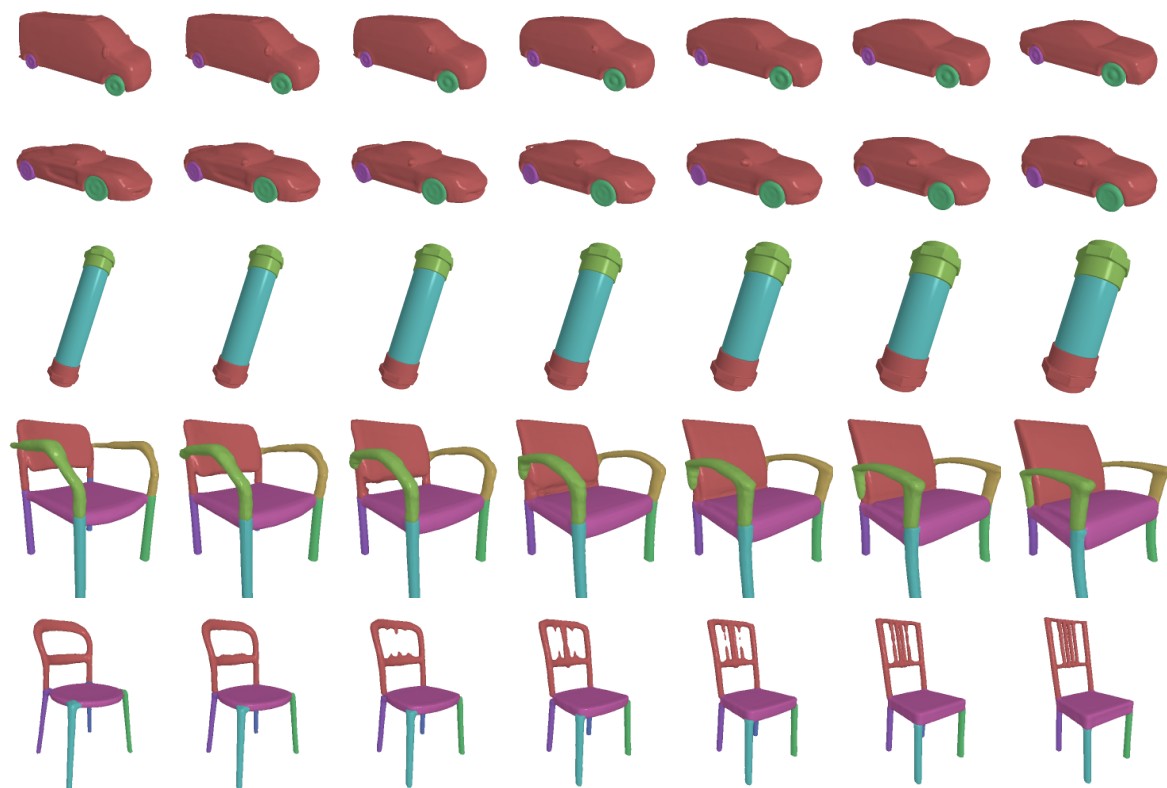

Figure 15: **Part interpolation.** On each row, we interpolate between the part latents and poses of one shape (*left*) to the other (*right*) and reconstruct the intermediate shapes. The overall shape structure is preserved while the parts change smoothly, remaining coherent with each others.

| Before | After |
|--------|-------|
| 0.378  | 0.325 |

with an average relative improvement of 12.7%. We visualize some optimization results in Figure 14. As can be observed, only the body of the car has been modified while the wheels are unchanged. Nonetheless, they stay consistent with each other and no inter-penetration happens.

## E   Part Interpolation

To demonstrate the smoothness and consistency of our latent and pose space, we perform an interpolation experiment. Given pair of shapes, we linearly interpolate between their part latent vectors and poses, with the exception of rotation quaternions for which we use spherical linear interpolation (slerp). At multiple interpolation steps, we reconstruct the intermediate shapes using our decoder and visualize such interpolation "path" in Figure 15, illustrating smooth transitions between the pair of shapes. Notably, the part-based structure remains coherent throughout the interpolation. This highlights the effectiveness of our learned latent and pose spaces in representing composite shapes.

## F   Results on DrivAerNet++

To evaluate PartSDF in a realistic industrial setting, we trained our model on DrivAerNet++ (Elrefaie et al., 2024), a high-fidelity automotive dataset designed for aerodynamic analysis and shape optimization. The

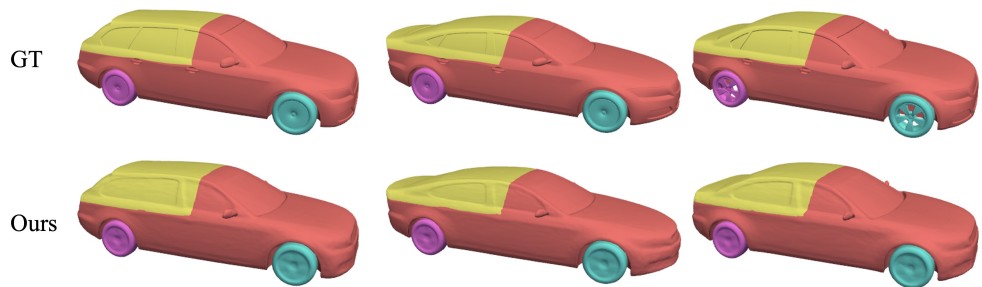

Figure 16: **Reconstructions on DrivAerNet++.** Reconstructions from the test set of the DrivAerNet++ dataset. PartSDF successfully captures fine geometric details and preserves consistency between parts.

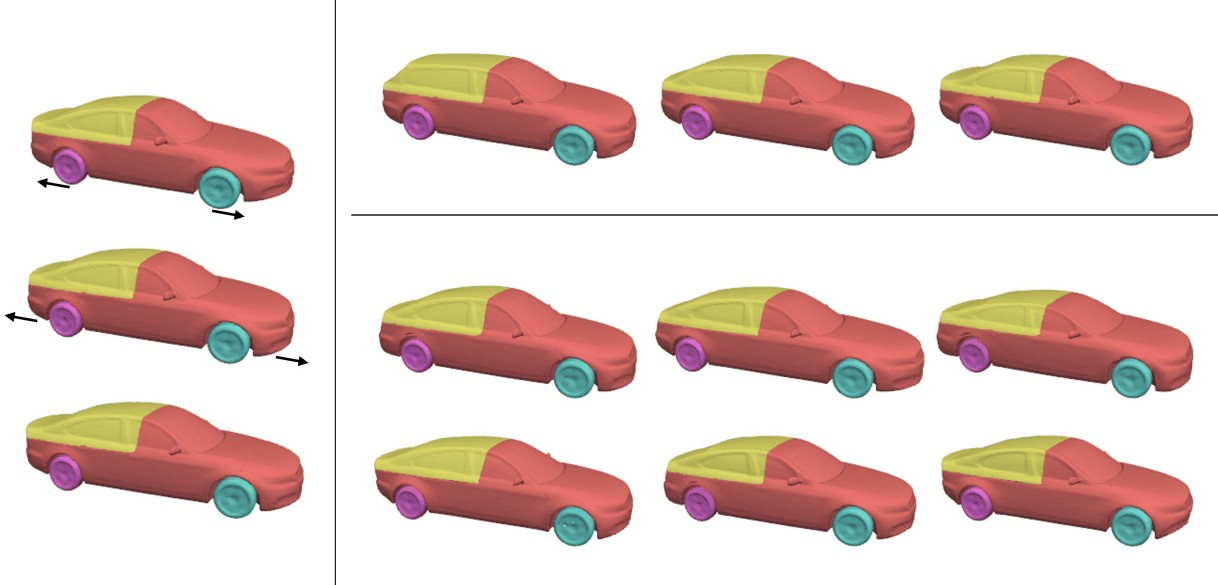

Figure 17: **Manipulation on DrivAerNet++.** (Left) Wheels are translated, then the car body is elongated. (Right-top) We interpolate from an Estateback to a Notchback in a smooth manner. (Right-bottom) We sample random body and back latents with the same wheels and poses to generate varying designs with similar general scale.

dataset includes three base car designs, each with thousands of geometric variations capturing diverse body, back, and wheel configurations.

We used these data to train PartSDF with separate parts for the car body, car back, and wheels, using the same architecture and loss functions as in our main experiments. Figure 16 shows reconstruction examples on held-out test shapes, demonstrating that PartSDF accurately captures fine geometric details and maintains part consistency on these industrial-grade models.

Furthermore, as illustrated in Figure 17, PartSDF enables smooth and coherent manipulation and interpolation between car designs. These results highlight the ability of PartSDF to generalize to high-fidelity shapes and support structured manipulation in practical aerodynamic design scenarios.

## G   Additional Experimental Results

In this section, we provide additional results and visualization on shape reconstruction (Appendix G.1), with new object categories, generation (Appendix G.2) and manipulation (Appendix G.3).

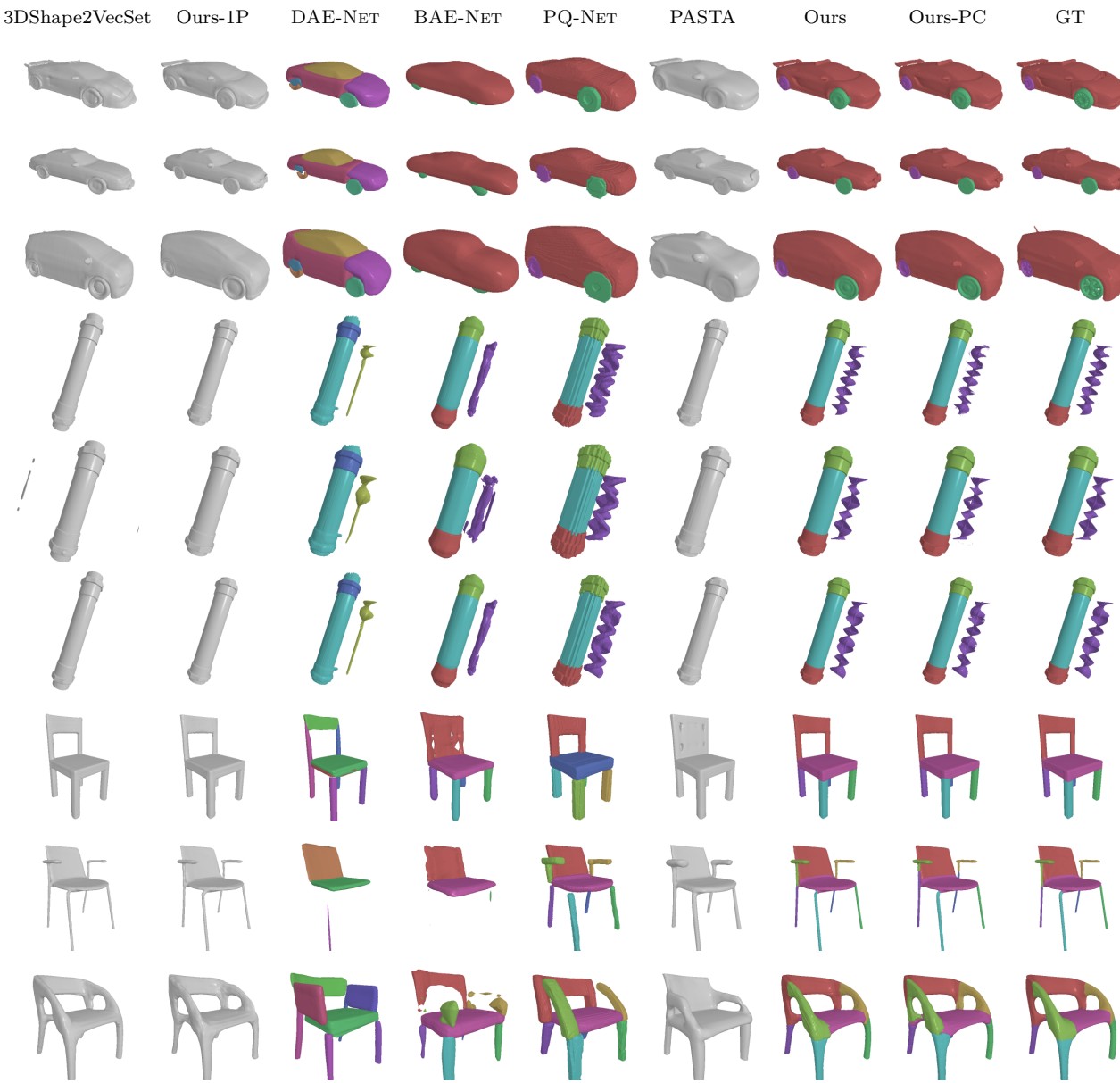

Figure 18: **Additional shape reconstruction.** Reconstructions of test shapes using the baselines and our model. We use a single part with Ours-1P and Ours-PC is the reconstruction using the point cloud encoder, as described in Section 4.1. For part-based method, we translate the helix outside of the mixers for visualization.

## G.1 Shape Reconstruction

We provide additional examples of shape reconstruction on the test set in Figure 18. We also visualize the part poses as primitives and results using the point encoder. DAE-NET and BAE-NET struggle to recover thin parts and complex shapes, while PQ-NET tend to inflates them without details. On the opposite, PartSDF recovers accurately the shapes and the parts. We note that with a point cloud encoder, the results may not be as accurate, *e.g.*, the last chair where the armrest segmentation is different, though it is still sensible.

Table 8: **Additional categories.** We report reconstruction metrics on test shapes for two new categories: *Display* and *Knife*.

| | CD (↓) | | IoU (%, ↑) | | IC (↑) | | pIoU (%, ↑) | |
|---|---|---|---|---|---|---|---|---|
| | Display | Knife | Display | Knife | Display | Knife | Display | Knife |
| 3DShape2VecSet | 4.95 | 1.72 | 92.57 | 80.33 | 0.949 | 0.870 | - | - |
| Ours-1P | 4.45 | 0.35 | 95.02 | 93.89 | 0.949 | 0.943 | - | - |
| DAE-Net | 98.75 | 489.58 | 58.67 | 46.63 | 0.772 | 0.567 | 45.76 | 42.39 |
| BAE-Net | 161.16 | 387.71 | 43.72 | 34.26 | 0.718 | 0.536 | 29.81 | 31.41 |
| PQ-Net | 95.73 | 41.60 | 42.81 | 30.45 | 0.775 | 0.560 | 41.44 | 36.09 |
| PASTA | 148.07 | 10.57 | 60.01 | 58.01 | 0.834 | 0.718 | - | - |
| Ours | **1.73** | **0.28** | **97.37** | **96.16** | **0.965** | **0.958** | **89.11** | **92.81** |

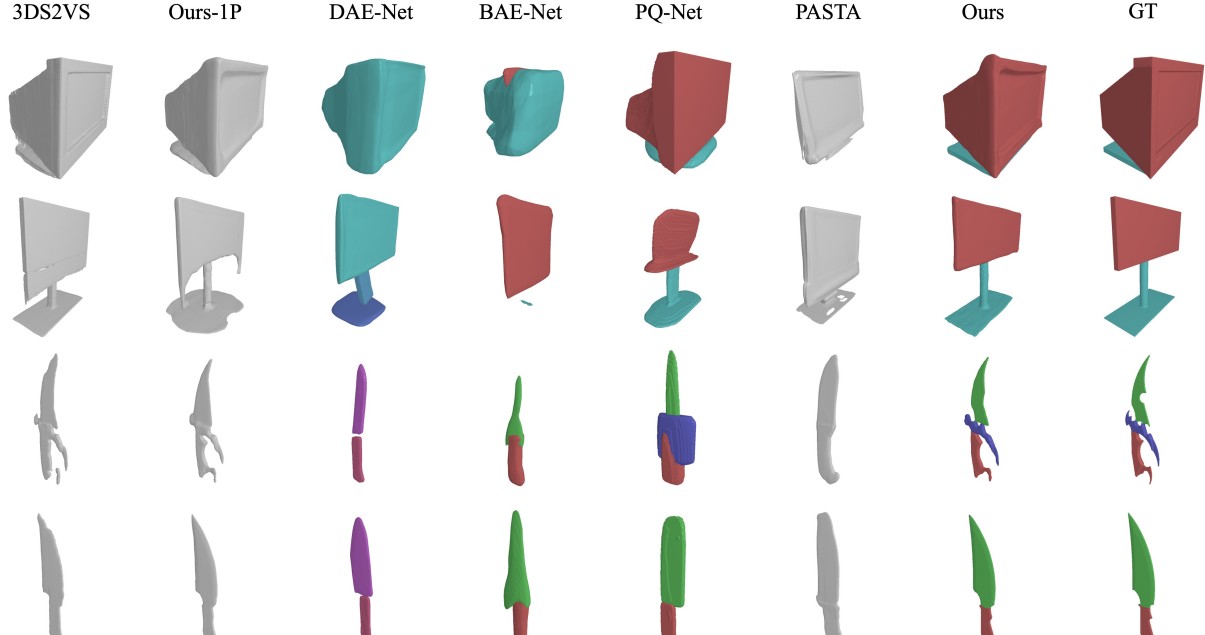

Figure 19: **Additional categories.** Reconstructions of test shapes using the baselines and our model on two new categories: *Display* and *Knife*.

Additionally, we expand our evaluation to additional categories from ShapeNet and PartNet: *Display* and *Knife*, featuring 640 and 324 shapes with 2 and 3 parts respectively. Results are given in Table 8 and Figure 19. They confirm that PartSDF delivers a strong performance beyond the original categories. In particular, the CD value for our method using multiple parts is markedly better on the *Displays*, mostly because baselines tend to fail drastically for a few shapes, whereas our complete approach succeeds. Voxel-based approaches DAE-Net, BAE-Net, and PQ-Net particularly struggle on thinner shapes, which can result in very large CDs. We also observe that PASTA struggles on all datasets because it only uses parts bounding box as information for reconstruction.

## G.2 Shape Generation

In Figure 20, we visualize generation examples with PQ-Net and PartSDF. We also show the part latent generation conditioned on poses for our model. The generated shapes with our method in combination of SALAD (Koo et al., 2023) are more detailed than PQ-Net + a latentGAN, with greater variety for cars and mixers (as reported in Table 2).

3DShape2VecSet

PQ-Net

PASTA

Ours

Ours†

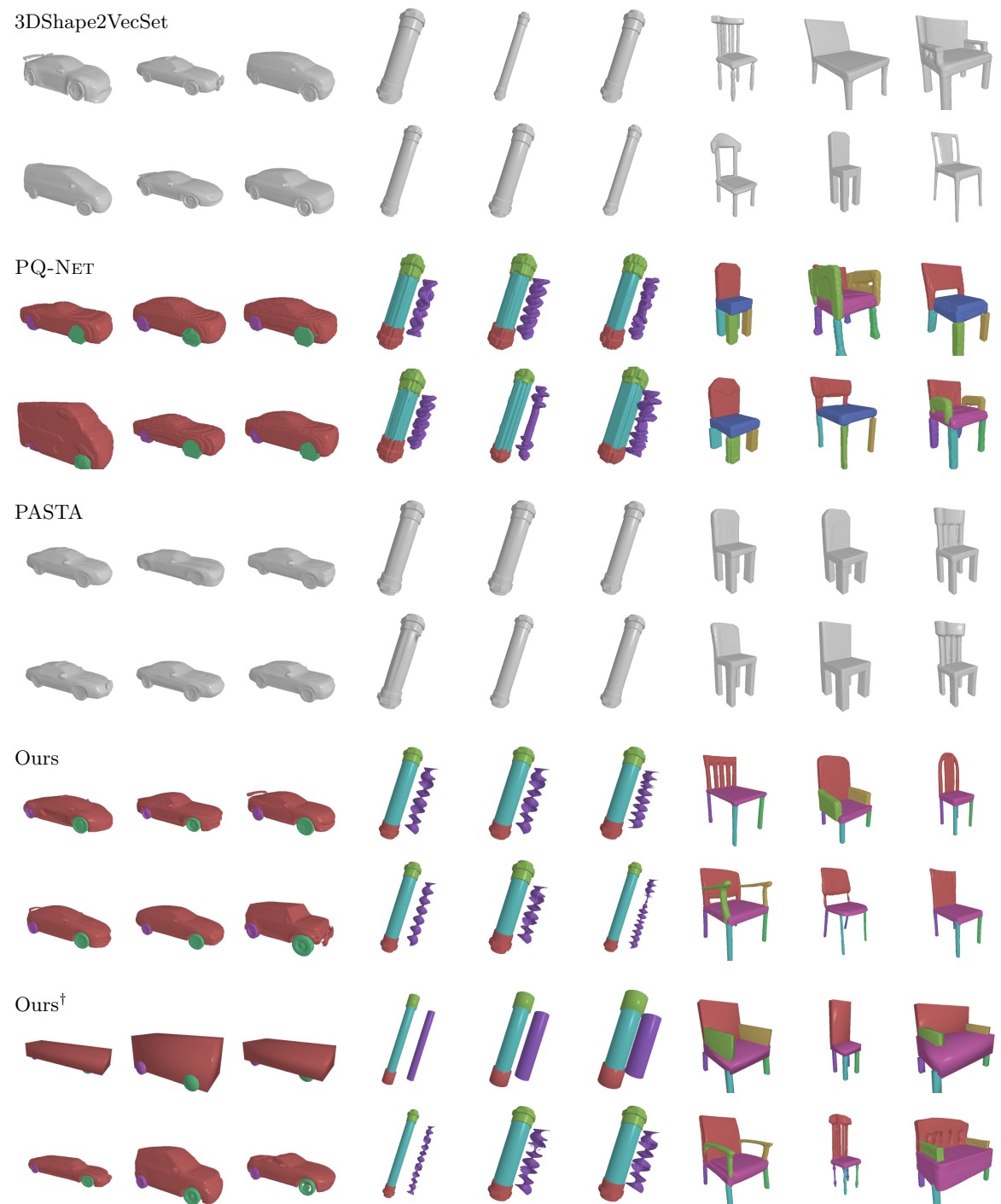

Figure 20: **Additional shape generation.** Randomly generated shapes and their parts on all datasets. We also provide examples of pose conditioned generation (Ours†) where part latents are generated based on the poses' coarse description of the shape (top image for each pair). When possible, the helix is translated outside of the mixers for visualization.

## G.3 Shape Manipulation

Finally, we perform more manipulation in Figure 21 with PartSDF on all dataset. By changing part latents, the affected parts have modified appearances but adapt to the original pose and to each others. For example,

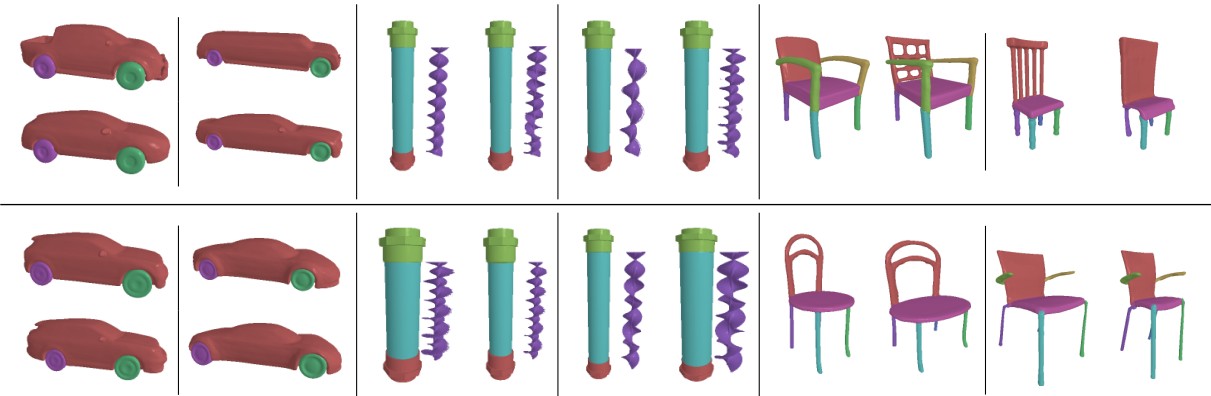

Figure 21: **Additional shape manipulation.** We manipulate four shapes per dataset: (*top*) by changing the latent of specific parts (car body, mixer helix, and all chair parts) and (*bottom*) by editing part poses (car wheels, mixer width, chair width and height). In all cases, the parts adapts to the modifications and to each other, maintaining a coherent whole.

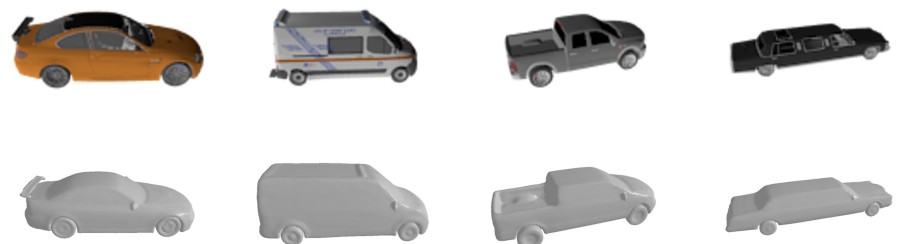

Figure 22: **Single-view reconstruction.** Given a single RGB image (top), a lightweight ResNet-18 encoder predicts part latents and poses for our pretrained part decoder, which reconstructs the composite 3D shape (bottom). The model recovers coherent part layouts and overall geometry.

the car bodies have the same general scales but their wheel wells fit correctly the original wheels, the mixers helices fit the original tube, and the chairs keep the same overall structure while each parts are locally different. When editing their poses, the parts are moved and deformed, but also preserve their original features, *e.g.*, the cars keep their original shapes but adapt to their new wheels, the mixers and their helices are rescaled but conserve their inside/outside relationship, and the size and height of chairs can be changed while maintaining the chairs' features. Note that this requires the part poses to be correct and coherent. Indeed, if they are manually set to be out of contact and disorganized in 3D space, the resulting shape will be meaningless.

## H   Single-View Reconstruction

While our main application focuses on shape optimization, we also experiment with single-view 3D reconstruction to demonstrate the feasibility of integrating PartSDF into an image-based pipeline. For this proof-of-concept, we trained a small image encoder to predict the part latents and poses corresponding to our pretrained part decoder. The encoder is a ResNet-18 model trained on synthetic renderings from ShapeNet (Chang et al., 2015) using the dataset of 3D-R2N2 (Choy et al., 2016a).

In Figure 22, we show reconstruction examples on held-out test images. Despite the simplicity of this setup and limited training data, the results demonstrate that PartSDF can, in principle, recover coherent part-aware 3D structures from single RGB images. Extending this to real-image datasets is a promising direction for future work.

