# OpenReview forum: "PartSDF: Part-Based Implicit Neural Representation for Composite 3D Shape Parametrization and Optimization"
_TMLR — Accepted by TMLR_

### Review · Reviewer_J4Dm · 2025-08-13

**Summary Of Contributions:**

This paper proposes PartSDF, a supervised implicit 3D shape representation that models objects as assemblies of independent, controllable parts while preserving global consistency. The method introduces a part-aware implicit architecture enabling precise per-component manipulation for engineering design workflows. The authors conduct detailed experiments, showing the state-of-the-art reconstruction and generation performance over supervised and unsupervised baselines.

**Audience:**

Yes

**Claims And Evidence:**

Yes

**Requested Changes:**

See [weaknesses] above.

**Strengths And Weaknesses:**

Strengths:
- The authors proposes modeling composite shapes using independent part-level signed distance functions (SDFs), thereby allowing precise control over individual components while maintaining overall coherence.
- The authors conduct detailed experiments and ablation studies, demonstrating superior performance in both reconstruction and generation tasks against supervised and unsupervised baselines.
- The paper is well-written and is easy to understand.
- The proposed single-decoder architecture is elegant and lightweight compared to more complex multi-step or post-processing methods.

Weaknesses:
- Experiments appear to focus on a narrow scope of the datasets. The generality of the method would benefit from tests on more diverse categories and real-world examples.
- While comparisons include supervised and unsupervised baselines, it misses the discussions with some recently related work such as PartGen (CVPR 2025).
- The paper would benefit from deeper analysis of design choices—e.g., how different numbers of parts, architectural variants, or SDF aggregation strategies affect reconstruction quality and control capability.
- It's unclear how the method handles objects with a very high number of parts. Questions remain about model size, training cost, and performance degradation when handling more complex objects.

---

> ### Author Response · Authors · 2025-09-15
> **Rebuttal 1**
>
> We thank the reviewer for their time and constructive feedback. Below we address the main concerns.
>
> ### 1. Dataset Scope and Generality
>
> We agree that broader evaluation would strengthen the work. Our initial choice of datasets (Cars, Mixers, Chairs) was intended to provide clean and consistent part decompositions that reflect a potential engineering use, while featuring interesting features and challenging structures (e.g., wheel-body separations in cars, thin mixer helices, multiple small chair parts).
>
> We are working on expanding our evaluation to additional categories from ShapeNet and PartNet. We have new results on the *Display* and *Knife* categories, featuring 640 and 324 shapes with 2 and 3 parts respectively. The tables and images below depict our results. They confirm that PartSDF delivers a strong performance beyond the original categories. In particular, the CD value for our method using multiple parts is markedly better on the *Displays*, mostly because baselines tend to fail drastically for a few shapes, whereas our complete approach succeeds.
>
> Voxel-based approaches DAE-NET, BAE-NET, and PQ-NET particularly struggle on thinner shapes, which can result in very large CDs. We also observe that PASTA struggles on all datasets because it only uses parts bounding box as information for reconstruction.
>
> Testing on more categories is a relatively slow process because it takes time to properly process all the data—to make all the meshes watertight, for instance—and to run our method and the baselines.
>
> | **Display** | CD $\downarrow$ | IoU (%) $\uparrow$ | IC $\uparrow$ | pIoU (%) $\uparrow$ |
> | --- | --- | --- | --- | --- |
> | 3DShape2VecSet | 4.95 | 92.57 | 0.949 | - |
> | Ours-1P | 4.45 | 95.02 | 0.949 | - |
> | DAE-NET | 98.75 | 58.67 | 0.772 | 45.76 |
> | BAE-NET | 161.16 | 43.72 | 0.718 | 29.81 |
> | PQ-NET | 95.73 | 42.81 | 0.775 | 41.44 |
> | PASTA | 148.07 | 60.01 | 0.834 | - |
> | Ours | 1.73 | 97.37 | 0.965 | 89.11 |
>
> | **Knife** | CD $\downarrow$ | IoU (%) $\uparrow$ | IC $\uparrow$ | pIoU (%) $\uparrow$ |
> | --- | --- | --- | --- | --- |
> | 3DShape2VecSet | 1.72 | 80.33 | 0.870 | - |
> | Ours-1P | 0.35 | 93.89 | 0.943 | - |
> | DAE-NET | 489.58 | 46.63 | 0.567 | 42.39 |
> | BAE-NET | 387.71 | 34.26 | 0.536 | 31.41 |
> | PQ-NET | 41.60 | 30.45 | 0.560 | 36.09 |
> | PASTA | 10.57 | 58.01 | 0.718 | - |
> | Ours | 0.28 | 96.16 | 0.958 | 92.81 |
>
> [Reconstruction results on Display and Knife](https://ibb.co/ymMKBmV7)
>
> Perhaps even more significantly, to provide results that mirror more closely a true industrial use case, we have been collaborating with an industrial partner on incorporating PartSDF into their design pipeline and testing it the result on DrivAerNet++ [1], a high-fidelity automotive dataset aimed at aerodynamic studies and design. This dataset features 3 cars designs with thousands of variations.  We trained PartSDF on it, with separate parts for the car body, car back, and wheels. In the figure and animation below, we show that PartSDF successfully handles these more realistic car geometries and delivers a smooth interpolation from one to the next. This shows that PartSDF works well on a true industrial-grade dataset, something that has not been demonstrated for the other baselines. And we will make the code publicly available.
>
> [Reconstruction on DrivAerNet++](https://ibb.co/rRTbrmCx)
>
> [Shape manipulation on DrivAerNet++](https://ibb.co/5xhH5BLK)
>
> ### 2. Updated Related Work
>
> We appreciate the pointer to this recent work, and we will add the following discussion in the revised manuscript.
>
> > Several recent works explore part-aware 3D generation, typically from images. Part123 [2] and PartGen [3] both begin by generating multi-view renderings with associated 2D part segmentations. Part123 then employs NeuS to recover part geometries, while PartGen predicts completed part images to fill occlusions before reconstructing their 3D geometry; it can also operate from text prompts in addition to images. A concurrent work, MIDI [4], addresses single-view scene reconstruction by segmenting objects and conditioning a 3D diffusion model on each object instance, with cross-object attention.
> > Unlike PartSDF, these approaches primarily focus on image-to-3D pipelines and emphasize image reconstruction, as opposed to geometric accuracy or ease of manipulation. In contrast, PartSDF learns supervised, part-aware implicit representations with explicit latent parametrization, enabling coherent part manipulation and constrained optimization, which are crucial in engineering and design scenarios

---

> ### Author Response · Authors · 2025-09-15
> **Rebuttal 2**
>
> ### 3. Design choices
>
> #### Number of parts
>
> We evaluated the influence of the number of parts on the Chair dataset by artificially fusing or splitting parts to vary the decomposition. We report the results in table below.  Increasing the number of parts generally improves reconstruction metrics, as simpler per-part geometries are easier to represent. However, we observe a slight Chamfer-Distance degradation due to a few failures in removing armrests during reconstruction, which suggests that larger part counts may require more data and training to robustly capture cross-part interactions. Part IoU remains relatively stable, with only minor variation ($\simeq0.6\%$). Overall, while finer decompositions can improve expressiveness, they may reduce controllability by requiring more poses to be specified and sufficient training data to model all part relationships.
>
> | **# of parts** | CD $\downarrow$ | IoU (%) $\uparrow$ | IC $\uparrow$ | pIoU (%) $\uparrow$ |
> | --- | --- | --- | --- | --- |
> | 1 | 3.74 | 93.78 | 0.913 | - |
> | 2 | 1.61 | 95.91 | 0.931 | 93.08 |
> | 3 | 2.00 | 96.55 | 0.936 | 93.24 |
> | 4 | 1.27 | 97.13 | 0.943 | 93.69 |
> | 5 | 1.51 | 97.23 | 0.943 | 93.63 |
> | 7 | 1.32 | 97.21 | 0.944 | 93.12 |
> | 8 | 1.30 | 97.17 | 0.942 | 93.67 |
> | 9 | 1.39 | 97.37 | 0.945 | 93.22 |
> | 12 | 1.76 | 97.48 | 0.945 | 93.01 |
>
> [Example of part decomposition for the Chairs](https://ibb.co/yFbs61PR)
>
> #### Architectural variants
>
> We experimented with a transformer-based architecture to replace our cross-part decoder with self-attention, see the table below. While feasible, this increases parameter count (from ~2.5M to 4.8M), memory (traing samples reduced by a factor 4 to avoid out-of-memory errors) and training cost (from 2000 to 4000 epochs with lower learning-rate) for the same reconstruction accuracy. We also observed greater parts entanglement in this architecture: displacing the car wheels also affect the car length even when its scale is kept constant, see the image below where the car front and back move with the wheels instead of simply adapting their wells around them. We believe this could be solved, for example by adding some shape mixing during training, but we note that our simpler cross-part decoder already enables it at a lower memory and computation cost.
>
> |  | CD $\downarrow$ | IoU (%) $\uparrow$ | IC $\uparrow$ | pIoU (%) $\uparrow$ |
> | --- | --- | --- | --- | --- |
> | Transf. | 1.25 | 98.00 | 0.931 | 93.44 |
> | Ours | 1.27 | 98.02 | 0.931 | 94.89 |
>
> [Comparison on manipulation](https://ibb.co/yF2VK9MJ)
>
> #### SDF aggregation strategies
>
> The merger of two SDFs into a single one is performed using the $\min$ operator, *e.g.* [5], which simply aggregates the parts. We tried  replacing the $\min$ by a $\mathrm{softmin}$ variant and found that it performs similarly, if marginally worse, as shown below for the Car dataset.
>
> |  | CD $\downarrow$ | IoU (%) $\uparrow$ | IC $\uparrow$ | pIoU (%) $\uparrow$ |
> | --- | --- | --- | --- | --- |
> | $\mathrm{softmin}$ | 1.31 | 97.90 | 0.930 | 94.78 |
> | $\min$ (Ours) | 1.27 | 98.02 | 0.931 | 94.89 |
>
> ### 4. Scalability of PartSDF
>
> The computational cost of PartSDF depends on both the number of parts $P$ and the feature dimension $D$. The single-part layers $h_\mathrm{sp}$ scale as $O(PD^2)$, while the cross-part layers $h_\mathrm{cp}$ scale as $O(P^2D)$. In our experiments, $D\in\{256, 512\}$ and $P\ll D$, so runtime and memory scale approximately linearly with the number of parts. However, for very large $P$, the combined complexity becomes $O(PD^2+P^2D)$, similar to the cost of transformer layers (considering self-attention + MLP). Although our layers remain more lightweight, modeling highly complex objects with many parts would require both large $D$ and $P$ to capture fine geometric details, which may limit scalability.  Extending our work to such cases is possible but will require strategies such as positional encoding, hierarchical part grouping, or sparse cross-part interactions to improve efficiency. We will clarify this limitation in the revised manuscript.
>
> ### References
> [1] DrivAerNet++, Elfraie et al., NeurIPS 2024
> [2] Part123, Liu et al, SIGGRAPH 2024
> [3] PartGen, Chen et al., CVPR 2025
> [4] MIDI, Huang et al., CVPR 2025
> [5] https://iquilezles.org/articles/distfunctions/, under "Primitive combinations"

---

### Review · Reviewer_wdYS · 2025-09-02

**Summary Of Contributions:**

This paper provides a deep neural model for 3D shapes that can be used for reconstruction, manipulation, and optimization -- with the main novelty in how it handles the decomposition of the overall shape in to separate parts.

The technical contribution is in the neural architecture, which has alternating layers between ones for the entire shape, and for representation by parts.  This allows for carefully structured loss functions.

There is a nice and thorough experimental section on shape datasets (each with decomposition into about 5 parts), for cars, chairs, and mixers.  It demonstrates the networks ability in auto-encoding, generative modeling, and manipulation, and optimization.  The optimization in particular, where a car body is optimized without changing the wheels, shows the power and promise of the part-decomposed modeling.

**Audience:**

Yes

**Claims And Evidence:**

Yes

**Requested Changes:**

At least report the relative size of the baseline models interns of memory or number of parameters.  If possible and they are smaller, scale up some baseline model to see if they match the performance of the PartSDF model.
Alternatively, you could show a version of PartSDF with fewer layers to shrink it in a way that would match baselines.

**Strengths And Weaknesses:**

Strengths:
  + Strong experimental section, with many uses from one architecture demonstrated, and ablation study on the loss function components.
  + Their method improves upon the baselines in almost all instances, including often improving upon reconstruction methods against baselines that do not also decompose into parts.
  + Compelling application (in one-part optimization) that well-motivates why this parts-based decomposition could be very useful.


Weaknesses:
  - I would like to see how the baseline models compared in size to the proposed PartsSDF model.  Is the improvement simply because the proposed model is larger and trained larger?  If, for instance, 3Dshape2VecSet model is smaller, but scaled up to match Ours-1P, would it perform as well?

---

> ### Author Response · Authors · 2025-09-15
> **Rebuttal**
>
> We thank the reviewer for their time and constructive feedback
>
> ### 1. Model size comparison
>
> We compare the model sizes in term of their number of trainable parameters. As can be seen in the table below, PartSDF is the smallest model, with 3DShape2VecSet being $\sim40\times$ larger. In practice, 3DShape2VecSet also requires multi-GPU training, as noted in its paper, while PartSDF can be trained on a single GPU.
>
> | | # of parameters |
> | --- | --- |
> | 3DShape2VecSet* | 106.1M  |
> | Ours-1P | 2.5M  |
> | DAE-NET | 3.1M  |
> | BAE-NET | 5.3M  |
> | PQ-NET | 12.8M  |
> | PASTA$^\dagger$ | 9.0M  |
> | Ours | 2.5M  |
>
> *: Most parameters are in the MLPs of the Self-Attention layers between the latent vectors.
> $^\dagger$: We only consider the parameters of the *blending network* and ignore the *object generator*, which is only relevant for generation.
>
> To understand the good performance of PartSDF in spite of its smaller size, please note that 3DShape2VecSet is an **auto-encoding** method designed for point cloud reconstruction and diffusion-based generation, not for per-shape optimization. In contrast, PartSDF is an **auto-decoding** framework that is well-suited for optimization and manipulation tasks. For completeness, we also experimented with optimizing 3DShape2VecSet latents at inference time on the *Car* dataset, with two settings: "+AD", where latents are randomly initialized and optimized, as in PartSDF, and "+AE+AD", where latents are first obtained from the point-cloud encoder and then further optimized. We report the results in the table below.
>
> |  | CD $\downarrow$ | IoU (%) $\uparrow$ | IC $\uparrow$ | pIoU (%) $\uparrow$ |
> | --- | --- | --- | --- | --- |
> | 3DShape2VecSet | 2.73 | 92.03 | 0.887 | -  |
> | 3DShape2VecSet+AD | 57.97 | 92.49 | 0.848 | -  |
> | 3DShape2VecSet+AE+AD | 1.15 | 98.78 | 0.942 | -  |
> | Ours-1P | 1.37 | 97.41 | 0.924 | -  |
> | Ours | 1.28 | 97.95 | 0.930 | 94.25 |
>
> In the +AD setting, 3DShape2VecSet struggles to optimize its latent vectors and the reconstructions exhibit bumpy surfaces and floaters, hence the lower IC and poor CD. This is consistent with observations made in the DeepSDF paper's supplementary material (Sec. E, Fig. 19). This highlights that its latent space is not well suited for direct optimization, one of the main reasons we did not adopt a VecSet-style latent space for PartSDF. With +AE+AD, performance improves and slightly surpasses that of PartSDF in reconstruction accuracy.
>
> However, 3DShape2VecSet is not a good fit for the applications we are targeting: It does not provide explicit part representations and its latent space is not designed for controllability or per-part optimization. To demonstrate this, we added small amounts of Gaussian noise to the latent vectors of 3DShape2VecSet and PartSDF before reconstruction. In the figure below, the standard deviation of the noise is taken to be 5\% ( +noise ) and 7.5\% ( ++noise ) of the latents average norm. Adding such small amounts of noise to 3DShape2VecSet's latent vectors  quickly degrades  the shape, while PartSDF's output remains a valid car. This is not an isolated example and happens for most cars in the dataset. In other words, our latent space is better behaved and, thus, better suited for shape optimization.
>
> [Reconstruction with Gaussian noise](https://ibb.co/dw7WsRZZ)

---

> > ### Comment · Reviewer_wdYS · 2025-09-15
> >
> > Thanks.  This clearly and satisfactorily addresses my main concern about the paper.

---

### Review · Reviewer_aHYc · 2025-09-08

**Summary Of Contributions:**

The paper introduces a supervised implicit representation framework for modeling composite 3D shapes with distinct, controllable parts. The core contributions are:

- A architecture that uses a shared implicit model to decode each part, represented by a latent vector and a pose. The full shape is formed by taking the minimum of the part SDFs, ensuring global coherence and seamless composition.
- A new supervision technique that bypasses the need for clean, watertight part meshes.
- Demonstrating the model's versatility as a shape prior for multiple tasks, including shape reconstruction, part-aware generation, and constrained optimization, all with the same core auto-decoder architecture.

**Audience:**

Yes

**Broader Impact Concerns:**

The reliance on pre-labeled data is noted as a limitation, which could be a concern for fairness and bias if the curated datasets do not represent a broad range of shapes or designs.

**Claims And Evidence:**

Yes

**Requested Changes:**

The following adjustments would strengthen the work in the reviewer's view, also including the weaknesses:

1. The authors should investigate developing more advanced topological supervision for each part to further improve robustness and prevent topologically inconsistent predictions. I still have some concerns about the ability of topological representation, especially for the interpolation and optimization (The start shape is so close to the target shape).

2. To handle parts that are pure surfaces (e.g., a car hood) that are not naturally handled by SDFs, the authors could explore using unsigned distance functions in future work.

**Strengths And Weaknesses:**

**Strengths**

- PartSDF consistently outperforms both supervised and unsupervised baselines in reconstruction and generation tasks. The method yields superior accuracy by training with samples directly from the original shape, allowing it to capture detailed part-specific geometry and structure.
- The framework provides a versatile, fully differentiable, and multi-part representation. It enables precise control over individual components, which is crucial for engineering applications like design and optimization. The same decoder network can be used for various downstream tasks without modification.
- The paper is well-written and well-organized. The loss terms are well defined for the whole network.


**Weaknesses**

- What about the time efficiency for generation or interpolation tasks?
- What about the topology change during interpolation, since the SDF is topology-friendly? And if there are some parts that disappear during interpolation, will there be some abnormal shape generated?
- For the supplementary (GIF), the presented cases are relatively simpler. Could you show some complex cases to demonstrate the SDF's ability?
- Recently, there have been some large models for the part-level generation, such as the CAST (cast: component-aligned 3d scene reconstruction from an RGB image). I am curious about the comparison and performance.
- From the presented results, the visual quality seems better to me. Why not demonstrate the ability by some practical application, such as the single-view reconstruction? I suggest adding this application and performing some comparison with the existing works.
- The paper presents a good shape of the methodology and evaluations. I still suggest the author add more evaluations to present the complete evaluations for readers, since I still have unknown points about the methods. I think the other reader still has the same questions.
- The limitations are very clear. The main limitation of PartsDF is that it requires predefined part labels for training. While this fits well with computer-aided design (CAD) workflows where parts are known, it limits the method's applicability to datasets without such decompositions. And for the open surface are also drawbacks. And during the discussion of the limitation, could you show some failure cases to incorporate the limitation?
- For the dataset, could you show some noisy, incomplete, or non-watertight shape? I want to know the model can handle which kind of noisy, incomplete, or non-watertight shapes. Is it over-claim?

---

> ### Author Response · Authors · 2025-09-15
> **Rebuttal 1**
>
> We thank the reviewer for their time and constructive feedback, and address their concerns below.
>
> ### 1. Time efficiency
>
> The generation and interpolation of latents and poses is external to our proposed decoder. It  can be done, for instance, by a diffusion model trained to generate the latents and poses of the training shapes, and its speed will depend on the chosen implementation. For interpolation, we perform a simple linear interpolation between the latents and poses of two shapes. For example, to interpolate between two latents $z_1$ and $z_2$ with a factor $t\in[0, 1]$, we compute:
> $$ z = z_1 * t + z_2 * (t - 1). $$
> Then, the generated or interpolated latents and poses are decoded by our Cross-Part Decoder, as for reconstruction.
>
> We provide a comparison of model sizes in our answer to reviewer **wdYS**.
>
> ### 2. Topology change and additional vizualizations
>
> Good question, most interpolations we observed were stable, as parts are explicitly represented and cross-part layers encourage consistency, but we can observe occasional artifacts, e.g., when parts appear or disappear as intermediate shapes are ill-defined in this case. We add additional interpolation results and provide a visualization of a failure case below, along with new results on the DrivAerNet++ [1] dataset in our answer to reviewer **J4Dm**.
>
> [Additional interpolation results](https://ibb.co/4gnH1wwn)
> [Failure case with appearing armrests](https://ibb.co/dwKKqj26)
>
> ### 3. Related work
>
> We will add a discussion of CAST in our updated related work, along with the additional references provided by reviewer **J4Dm**. In short, CAST targets open-vocabulary scene reconstruction from a single RGB image, using large-scale pre-trained foundation models and 3D diffusion models ($\simeq1.5B$ parameters), as well as physics-aware post-processing. In contrast, PartSDF is a compact supervised framework (2.5M parameters) trained on part-annotated 3D datasets, to provide part-aware implicit representations that support per-part manipulation and optimization. The two approaches therefore address different needs: CAST is good for large-scale scene generation from images, while PartSDF provides lightweight, structured shape priors for controllable design workflows. We also note that CAST code and models are not yet public, so direct comparison is currently not feasible.
>
> ### 4. Single-view reconstruction
>
> We agree that single-image reconstruction is an interesting and valuable application, and we were planning to explore this as an extension of the present work in greater depth. In this paper, we use shape optimization as a practical application of PartSDF that highlights the benefits of explicit part-aware priors. Nevertheless, to demonstrate feasibility within the short time-window allowed by TMLR, we trained a small proof-of-concept image encoder to predict part latents and poses. We use a ResNet18 model and the renderings of ShapeNet from 3D-R2N2 [2]. We show examples of reconstruction on the test set below. They show that PartSDF can in principle be integrated into a single-view pipeline. However, establishing a comprehensive benchmark with real images and properly comparing to baselines and prior single-view methods will require a significant amount of time, which we do not have in the context of a rebuttal. We hope the reviewer will agree that it really should be its own paper.
>
> [Single-view reconstruction results](https://ibb.co/99HNNK0q)
>
> ### 5. Failure cases
>
> We will add visual examples of failure cases to the limitations section, shown in the image below. (*left*) When interpolating between shapes where a part is present in one but absent in the other, floating artifacts may appear (*e.g.*, chair armrests). (*middle*) In rare test cases, parts that should be absent may persist as very small surfaces, such as the small green armrest patches, which lead to higher Chamfer-Distances. In practice, this can be mitigated by simply ignoring the corresponding SDF if this part’s absence is known *a priori*, since PartSDF outputs parts individually. (*right*) Very thin structures, such as the helix of some mixers, are still challenging to represent and mesh reliably with an SDF-based approach.
>
> [Failure cases](https://ibb.co/wNHyrkCs)
>
> ### 6. Data robustness
>
> Our supervision strategy naturally accommodates non-watertight parts, *e.g.*, the chair of Figure 4, and we have included more examples to illustrate this below. For noisier or less complete parts, however, the supervision signal itself degrades, which in turn limits the quality of the learned shape prior. We acknowledge that our current formulation is not designed to explicitly denoise such data, and we will downplay any overclaims in the manuscript.
>
> [Non-watertight parts](https://ibb.co/0jYYVpJ2) (Note: parts are shown in an exploded view.)

---

> ### Author Response · Authors · 2025-09-15
> **Rebuttal 2**
>
> ### 7. Future work
>
> We agree that topological consistency and handling pure surface parts, such as car hoods, are important directions for future work, which we briefly discuss in the limitations and conclusion sections. Exploring topological supervision within and between parts might help prevent topologically incorrect shapes during manipulation, such as may happen with disappearing chair armrests, as above. On the other hand, unsigned distance functions have the potential to represent parts that are not volumetric, like car hoods, but are more challenging when it comes to recover a clean mesh from the implicit field.
>
> ### 8. Datasets bias and fairness
>
> We acknowledge that our curated datasets may not cover all shape categories, but they align with our focus on engineering and design contexts. To demonstrate broader applicability, we include results on additional categories and on DrivAerNet++ [1] in our answer to reviewer **J4Dm**.
>
> ### References
> [1] DrivAerNet++, Elfraie et al., NeurIPS 2024
> [2] 3D-R2N2, Choy et al., ECCV 2016

---

> ### Comment · Reviewer_aHYc · 2025-09-22
>
> Thanks for the great efforts! After reading the rebuttal, I am clear for the submission. The proposed method still suffers from the artifacts during the interpolation between the significant changes of topology. I suggest the authors make them clear in the paper and discuss the limitations well. For the data robustness part, the non-watertight shape, I am clear for the part without connection; instead of the more general shape (such as cloth and open surface), it should be more clear in the paper or show some cases for the cloth and open surface reconstruction.

---

### Decision · Action_Editor_4Hqa · 2025-10-06

**Recommendation:** Accept as is

**Audience:**

Yes

**Audience Explanation:**

The relation to representation learning is clear.

**Claims And Evidence:**

Yes

**Claims Explanation:**

All reviewers agree that the paper makes interesting and relevant contributions, and explains sufficiently well its positioning compared to other works. There are still criticisms regarding the part-to-part relations. Also, I would say that the paper relies on strong supervision for the final outputs and thus is not learning representations in unsupervised way that could easily scale to different object categories. As such, I do not recommend the journal-to-conference track.